# Differentially Private Preference Data Synthesis for Large Language Model Alignment

**Fengyu Gao** [1]  **Jing Yang** [1 2]

## Abstract

Preference alignment is a crucial post-training step for large language models (LLMs) to ensure their outputs align with human values. However, post-training on real human preference data raises privacy concerns, as these datasets often contain sensitive user prompts and human judgments. To address this, we propose **DPPrefSyn**, a novel algorithm for generating differentially private (DP) synthetic preference data to enable privacy-preserving preference alignment. DPPrefSyn is a principled framework grounded in the Bradley–Terry preference model and the intrinsic geometric structure of pairwise human preference data. It first learns an underlying preference model from private data with formal differential privacy guarantees, and then leverages the learned model together with public prompts to synthesize high-quality preference data. It exploits the shared linear structure of per-cluster reward models to effectively capture heterogeneous human preferences in private datasets, and leverages DP Principal Component Analysis (DP-PCA) to improve learning accuracy. Extensive experimental results demonstrate that DPPrefSyn achieves competitive alignment performance under strong DP guarantees. These findings highlight the potential of synthetic preference data as a practical alternative for privacy-preserving preference alignment across a broad range of applications. To the best of our knowledge, this is the first work to generate DP synthetic preference data for LLM alignment. Our code is available at https://github.com/gfengyu/Differentially-Private-Preference-Data-Synthesis.

## 1. Introduction

Preference alignment techniques, such as RLHF (Stiennon et al., 2020) and DPO (Rafailov et al., 2024), are widely used to align LLMs with human expectations. These algorithms rely on pairwise preference data, where human annotators compare two responses and select the one that better addresses the given prompt. Such preference data is then used to fine-tune LLMs by encouraging the model to rank preferred responses over less preferred ones (Ziegler et al., 2019; Ouyang et al., 2022; Bai et al., 2022). The effectiveness and reliability of preference alignment have motivated its adoption in applications such as chat assistants (Achiam et al., 2023), mathematical reasoning tools (Shao et al., 2024), and code generators (Shen et al., 2023).

However, aligning LLMs with human preference data raises substantial privacy concerns, since these datasets frequently include real user prompts and human feedback. These prompts may disclose personal information related to health, identity, or other sensitive topics, and human feedback may reveal private beliefs, preferences, or behavioral patterns (Li et al., 2023; Yu et al., 2024). Existing works (Chowdhury et al., 2024; Yu et al., 2024; Wu et al., 2023a) have explored approaches to mitigate these risks using the rigorous privacy safeguards provided by differential privacy (DP) (Dwork et al., 2006b) and shown encouraging results. However, many of these approaches protect only part of the data, for example, by privatizing either the user prompts (Yu et al., 2024) or the preference labels (Chowdhury et al., 2024; Zhang et al., 2025), but not both. Such partial protection leaves potential privacy gaps, as adversaries could still infer sensitive attributes from unprotected prompts or preference annotations. Besides, such works usually focus on specific post-training algorithms, such as RLHF (Wu et al., 2023a), and are not compatible with newer methods like DPO. Moreover, these methods are typically constrained by limited private preference datasets, and often fall short in achieving high-quality preference alignment, due to the high cost and scalability challenges associated with collecting human preference annotations. These gaps lead to the central question of this work:

*Is it possible to achieve superior preference alignment while ensuring thorough privacy protection and maintaining*

[1]Department of Computer Science, University of Virginia, Charlottesville, Virginia, USA [2]Department of Electrical and Computer Engineering, University of Virginia, Charlottesville, Virginia, USA. Correspondence to: Jing Yang <yangjing@virginia.edu>.

*Proceedings of the $43^{rd}$ International Conference on Machine Learning*, Seoul, South Korea. PMLR 306, 2026. Copyright 2026 by the author(s).

*compatibility with emerging alignment methodologies?*

In this work, we provide an affirmative answer to this question. Our main contributions are summarized as follows:

- We introduce a new algorithm DPPrefSyn (Figure 1) to generate DP synthetic preference data for LLM alignment. DPPrefSyn is a principled framework grounded in the Bradley–Terry preference model and the intrinsic geometric structure of pairwise human preference data. It features the following design elements: 1) *Capturing diverse human preferences via shared linear structure of rewards and clustering.* The Bradley–Terry preference model, together with a linear reward structure, enables us to group preference samples based on embedding differences between concatenated prompt–preferred and prompt–dispreferred pairs, and to approximate preferences within each cluster using a shared reward function. 2) *Improving learning efficiency through DP-PCA.* To address sample inefficiency and clustering instability in high-dimensional spaces, we apply DP-PCA to project the embeddings into a lower-dimensional space while preserving key preference signals under DP. 3) *Saving privacy budgets by using public prompts.* We use public prompts to generate candidate responses and apply private reward models to select preference pairs. This allows us to focus the privacy budget entirely on modeling private preferences, improving privacy efficiency.

- We conduct a rigorous privacy analysis of DPPrefSyn and confirm it follows $(\varepsilon, \delta)$-DP. In our analysis, we first allocate a privacy budget $\varepsilon_0$ to DP-PCA (Amin et al., 2019) for dimensionality reduction, and $\varepsilon_1$ to DP-KMeans (Su et al., 2016) for clustering. The remaining budget is used to train per-cluster reward models with DP-SGD. We apply the Privacy Random Variable (PRV) accountant (Gopi et al., 2021) to tightly compose the DP-SGD privacy cost, ensuring that DPPrefSyn satisfies $(\varepsilon, \delta)$-DP. Thanks to the post-processing property of DP, the resulting DP synthetic dataset can be reused across multiple preference alignment methods and different LLMs.

- We empirically evaluate DPPrefSyn on standard benchmarks, including question answering tasks from OpenAssistant (Köpf et al., 2023) and Anthropic-HH (Bai et al., 2022), as well as the TL;DR summarization task (Stiennon et al., 2020). Our experiments suggest that synthetic preference data generated by DPPrefSyn offers strong privacy guarantees while achieving competitive utility across diverse tasks. For example, with $\varepsilon = 2$, fine-tuning the Pythia-2.8B model using DPO on our synthetic data achieves a GPT-4o win rate of $56.48\%$ on Anthropic-HH, significantly outperforming fine-tuning on real data ($37.02\%$). These results show that our DP synthetic preference data can enable effective preference alignment of

LLMs with strong privacy guarantees. Moreover, DPPrefSyn can be applied to various preference optimization methods (e.g., DPO and RLHF) and remains effective under strong privacy constraints, overcoming the limitations of other privacy-preserving post-training approaches that are tailored to specific methods.

## 2. Related Work

**Differentially Private Alignment of LLMs.** Wu et al. (2023a) introduce a DP framework to align LLMs with reinforcement learning by adapting the PPO algorithm to the DP setting. However, their approach is *limited to PPO*, while preference alignment continues to evolve with new methods. Chowdhury et al. (2024) study the problem of reward estimation from preference-based feedback, using the notion of label-DP (Ghazi et al., 2021) to protect the privacy of human annotators. Zhang et al. (2025) propose AUP-RLHF, a user-level label-DP framework for RLHF. While both methods are effective at safeguarding preference labels, they *do not address the sensitivity of prompts or responses themselves.* Besides, such methods are usually *constrained by the limited private user preference data*, and may not achieve high-quality preference alignment, as human preference annotations are costly to obtain. Yu et al. (2024) focus on protecting sensitive user *instructions* by generating DP synthetic *instructions* to replace real ones during data annotation and model fine-tuning.

Beyond preference alignment, several studies focus on private fine-tuning of LLMs using general labeled data, rather than preference comparisons (Yu et al., 2021; Li et al., 2021; Chen et al., 2024; Tang et al., 2024; Zhang et al., 2023; He et al., 2022), and some recent work focuses on protecting the privacy of in-context prompts for in-context learning (Duan et al., 2023; Tang et al., 2023; Gao et al., 2024; Hong et al., 2023; Wu et al., 2023b).

**Differentially Private Synthetic Text Generation.** Our work falls within the broader scope of DP synthetic text generation. In this area, two representative lines of work have emerged. The first line fine-tunes a language model on private data under DP, and then uses the fine-tuned model to generate synthetic text (Yue et al., 2023; Mattern et al., 2022; Mireshghallah et al., 2023; Carranza et al., 2023; Yu et al., 2024; Wang et al., 2024; Ochs & Habernal, 2024; Tan et al., 2025; Carranza et al., 2024). For example, Yu et al. (2024) privately fine-tune the LLaMA model on private instruction data, and then use the resulting model to generate synthetic text for supervised instruction tuning.

Since DP fine-tuning can be expensive and sometimes infeasible (especially for non-public language models), a recent line of work explores DP synthetic data generation by prompting LLM APIs (Lin et al., 2023; Xie et al., 2024;

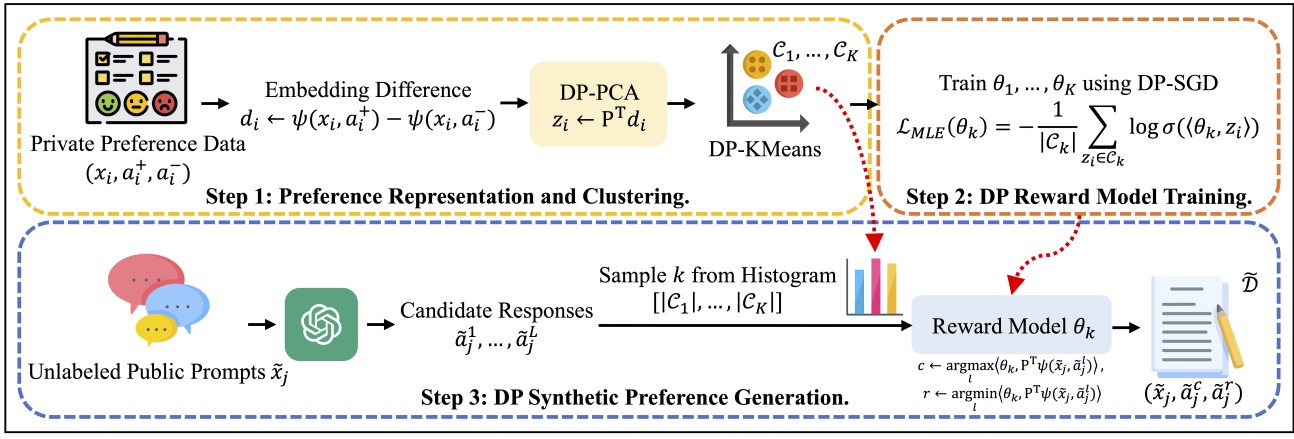

*Figure 1.* Overview of DPPrefSyn. DPPrefSyn generates DP synthetic preference data through 3 steps: 1) representing preference samples as embedding differences and clustering them via DP-PCA and DP-KMeans, 2) training DP reward models on each cluster using DP-SGD, and 3) generating synthetic preference samples from public prompts guided by the DP-protected distribution of reward functions.

Wu et al., 2024; Hou et al., 2024). These approaches typically generate synthetic samples using LLM API access and then select outputs that are similar to the private data in a privacy-preserving manner. For instance, Xie et al. (2024) propose the Aug-PE algorithm, which iteratively refines LLM-generated samples based on embedding-space similarity to private data. *Our work aligns with this line of research and focuses on preference alignment for LLMs.*

**Diversity in Human Preferences.** Recent works highlight that human preferences are inherently diverse (Denton et al., 2021; Aroyo et al., 2023a;b; Chakraborty et al., 2024). The key factors include sociodemographic differences (e.g., race, gender, age), personal beliefs and biases, varying levels of domain expertise, and the inherent ambiguity of natural language (Sandri et al., 2023; Vogels, 2021). Motivated by this, recent studies have explored personalizing preference alignment to better reflect the values of different user groups (Lee et al., 2024b; Poddar et al., 2024; Singh et al., 2025). For example, Singh et al. (2025) propose a meta-learning framework where an LLM adapts to individual users using a small number of preference examples. In this work, we generate DP synthetic preference data that *captures the underlying diversity of human feedback*, leading to improved alignment performance compared to approaches that ignore such diversity (as shown in Section 5.3).

## 3. Problem Definition

**Preference Alignment of LLMs.** We consider preference alignment of an LLM based on pairwise human feedback, using a preference dataset $\mathcal{D} = \{(x_i, a_i^+, a_i^-)\}_{i=1}^n$. Each sample consists of a prompt $x_i$ and two responses $a_i^+, a_i^-$, where $a_i^+$ is preferred over $a_i^-$ by human annotators.

Following prior works on reinforcement learning from human feedback (RLHF) and direct preference optimization

(DPO) (Ouyang et al., 2022; Zhu et al., 2023; Rafailov et al., 2024; Liu et al., 2023), we model human preferences using the Bradley–Terry model (Bradley & Terry, 1952):

$$\mathbb{P}[a_i^+ \succ a_i^- \mid x_i] = \frac{\exp(r^\star(x_i, a_i^+))}{\exp(r^\star(x_i, a_i^+)) + \exp(r^\star(x_i, a_i^-))}$$
$$= \sigma(r^\star(x_i, a_i^+) - r^\star(x_i, a_i^-)),$$

where $a_i^+ \succ a_i^-$ means $a_i^+$ is preferred to $a_i^-$, $r^\star(x, a)$ is a latent reward function, and $\sigma(z) = 1/(1 + e^{-z})$ is the sigmoid function.

This formulation underlies both RLHF, which explicitly learns a reward model from preference data, and DPO, which induces an implicit reward through the log-ratio between the learned policy and a reference policy.

**Differentially Private Preference Data Synthesis.** We aim to protect the privacy of the preference dataset $\mathcal{D}_{\text{priv}} = \{(x_i, a_i^+, a_i^-)\}_{i=1}^n$ against an adversary who attempts to access or infer private information about individual prompts, responses, or preference labels based on outputs of the fine-tuned LLM. To achieve this, we propose to generate a DP synthetic preference dataset $\tilde{\mathcal{D}} = \{(\tilde{x}_j, \tilde{a}_j^+, \tilde{a}_j^-)\}_{j=1}^m$ that preserves critical preference information in $\mathcal{D}_{\text{priv}}$. Thanks to the post-processing property of DP, $\tilde{\mathcal{D}}$ can be reused across various preference alignment methods and LLMs without incurring additional privacy cost.

We formally define $(\varepsilon, \delta)$-differential privacy as follows.

**Definition 3.1** ($(\varepsilon, \delta)$-Differential Privacy (Dwork et al., 2006a))**.** A randomized algorithm $\mathcal{A}$ is $(\varepsilon, \delta)$-differentially private if for any two neighboring inputs $\mathcal{D}$ and $\mathcal{D}'$ that differ by a single entry and any set $\mathcal{S}$ of possible outputs: $\mathbb{P}[\mathcal{A}(\mathcal{D}) \in \mathcal{S}] \leq e^\varepsilon \mathbb{P}[\mathcal{A}(\mathcal{D}') \in \mathcal{S}] + \delta$.

**Diverse User Preferences and Linear Reward Structure.** Human preferences in the private dataset are usually not

homogeneous (Chakraborty et al., 2024; Bakker et al., 2022; Kovač et al., 2023; Jang et al., 2023; Rame et al., 2023; Ji et al., 2023). Different annotators may prioritize different aspects of a response, such as factual accuracy, politeness, creativity, or clarity, resulting in diverse and sometimes conflicting signals in the data. To synthesize $\tilde{\mathcal{D}}$, it is thus critical to capture the diverse human preferences in $\mathcal{D}_{\text{priv}}$.

To facilitate this, we assume a linear structure for the latent reward function (Saha et al., 2023; Kong & Yang, 2022; Zhu et al., 2023). Specifically, we adopt a linear reward model

$$r_\theta(x, a) = \langle \theta, \phi(x, a) \rangle,$$

where $\phi(x, a) \in \mathbb{R}^d$ is the *shared* feature vector and $\theta \in \mathbb{R}^d$ is a *user-dependent* parameter.

The linear structure offers a balance between *expressiveness* and *tractability* for modeling diverse user preferences. While individual users may exhibit heterogeneous preferences, their reward functions often share common semantic features in a learned embedding space (i.e., $\phi(x, a)$), differing primarily in how these features are weighted (i.e., $\theta$). Such a linear parameterization captures this shared structure while allowing variability across users or preference clusters, enabling effective modeling of preference heterogeneity with limited data and strong privacy constraints. As demonstrated by our experimental results in Section B.6, the linear reward structure effectively captures preference signals and achieves competitive performance compared to a fully fine-tuned GPT-2 reward model, despite its simplicity.

## 4. Proposed Method

### 4.1. Key Components of the Algorithm Design

Before detailing our design, we highlight several key components of the algorithm that address the major challenges in synthesizing differentially private preference data.

**Leveraging Linear Reward Structure for Preference Clustering.** The linear reward model indicates that

$$\mathbb{P}[a_i^+ \succ a_i^- \mid x_i] = \sigma(\langle \theta, \phi(x_i, a_i^+) - \phi(x_i, a_i^-) \rangle).$$

Therefore, for a tuple $(x_i, a_i^+, a_i^-)$, $a_i^+ \succ a_i^-$ implies that $\langle \theta, \phi(x_i, a_i^+) - \phi(x_i, a_i^-) \rangle \gg 0$, i.e., the preference vector $\theta$ is strongly aligned with $\phi(x_i, a_i^+) - \phi(x_i, a_i^-)$. For human annotators with similar preferences, the feature difference vectors $\phi(x_i, a_i^+) - \phi(x_i, a_i^-)$ across their rated pairs are therefore expected to point in similar directions in the embedding space. Motivated by this intuition, we cluster the user preference samples based on the feature difference vectors $\phi(x_i, a_i^+) - \phi(x_i, a_i^-)$, such that user preferences within the same cluster can be well approximated by a shared reward function parameterized by a common preference vector $\theta$. In Section B.9, we show that the clusters constructed using this approach are meaningful and interpretable in terms of user preference styles.

**Adopting DP-PCA to Obtain Effective Features.** One natural choice for the feature vector $\phi(x, a)$ is the embedding of the corresponding (prompt, response) pair, as embeddings encode semantic similarities and differences that directly underlie human preference judgments. However, the embeddings are usually high-dimensional (e.g., 1024 dimensions) in order to capture rich semantic and stylistic features. Learning in such a high-dimensional space typically requires a large number of samples, which is inherently limited by the number of samples available within each cluster. To mitigate this issue, we apply differentially private PCA (DP-PCA) (Amin et al., 2019; Liu et al., 2022) to reduce the dimensionality of the embeddings and obtain compact feature representations for clustering and reward model training. This step improves sample efficiency and clustering stability while preserving the essential preference signals necessary for effective alignment.

**Utilizing Public Prompts to Save DP Budget.** Synthesizing prompts, preferred responses and less preferred responses simultaneously can quickly exhaust the privacy budget and degrade utility. To address this, we choose to use public prompts to avoid spending privacy budget on synthesizing DP prompts, allowing us to allocate the entire budget to modeling preferences over responses. For each public prompt, we generate multiple candidate responses using an LLM and apply a distribution of reward functions, which is trained from private preference data under DP, to construct preference pairs. While public prompts may differ in distribution from private ones, our experiments show that, despite this distribution shift in prompts, the synthetic preference data preserves the statistical preference distribution in the private preference dataset and remains effective for downstream preference alignment tasks.

### 4.2. DPPrefSyn Algorithm

We now introduce the DPPrefSyn algorithm (Algorithm 1), which generates a DP synthetic preference dataset from the private dataset $\mathcal{D}_{\text{priv}}$. DPPrefSyn consists of 3 main steps:

**Step 1: Preference Representation and Clustering.** We encode each prompt-response pair in $\mathcal{D}_{\text{priv}}$ using a public sentence embedding model $\psi$. Specifically, for each sample $(x_i, a_i^+, a_i^-)$, we first concatenate the prompt with each response to form two texts: $[x_i; a_i^+]$ and $[x_i; a_i^-]$. We then compute their embeddings via $\psi$, and define their difference as $d_i \leftarrow \psi(x_i, a_i^+) - \psi(x_i, a_i^-)$ (Line 3).

To reduce dimensionality privately, we apply DP-PCA (Amin et al., 2019) to $\{d_i\}_{i=1}^n$, obtaining a projection matrix $\mathbf{P} \in \mathbb{R}^{d \times p}$ under $\varepsilon_0$-DP (Line 5). Roughly speaking, DP-PCA (Amin et al., 2019) approximates the eigendecom-

position of the data covariance matrix by estimating the collections of eigenvalues and eigenvectors separately in a DP manner. Each $d_i$ is then projected to a lower-dimensional space as $z_i \leftarrow \mathbf{P}^\top d_i$ (Line 6). We apply DP-KMeans clustering (Su et al., 2016) on $\{z_i\}_{i=1}^n$ to group samples with similar preference patterns under $\varepsilon_1$-DP, forming $K$ clusters $\mathcal{C}_1, \ldots, \mathcal{C}_K$ in Line 7. Roughly, DP-KMeans (Su et al., 2016) is a DP version of the Lloyd algorithm.

**Step 2: DP Reward Model Training.** For each cluster $\mathcal{C}_k$, we learn the shared preference vector $\theta_k \in \mathbb{R}^p$ that captures preference patterns by minimizing the negative log-likelihood (Line 9):

$$\mathcal{L}_{\text{MLE}}(\theta_k) = -\frac{1}{|\mathcal{C}_k|} \sum_{z_i \in \mathcal{C}_k} \log \sigma\left(\langle \theta_k, z_i \rangle\right), \qquad (1)$$

where $z_i$ is the DP-PCA projected embedding difference from Step 1. Each reward model is trained with the DP-SGD algorithm (a DP variant of SGD) (Abadi et al., 2016) using noise multiplier $\sigma_2$.

**Step 3: DP Synthetic Preference Generation.** We compute a histogram $\mathbf{h} = [|\mathcal{C}_1|, \ldots, |\mathcal{C}_K|]$ representing the number of samples in each cluster (Line 11). We normalize it to get a probability distribution $\mathbf{p}$ over the clusters, i.e., $\mathbf{p} \leftarrow \mathbf{h}/|\mathcal{D}_{\text{priv}}|$ (Line 12).

Next, for each public prompt $\tilde{x}_j$, we use a relatively high temperature to prompt an LLM to generate $L$ diverse candidate responses $\tilde{a}_j^1, \ldots, \tilde{a}_j^L$ (Line 14). We then sample a cluster index $k \sim \mathbf{p}$ according to the DP histogram (Line 15) and evaluate the responses $\tilde{a}_j^1, \ldots, \tilde{a}_j^L$ using the reward model parameterized by $\theta_k$. To do this, we first compute the embeddings $\psi(\tilde{x}_j, \tilde{a}_j^l)$ for $l = 1, \ldots, L$, and project them into the DP-PCA subspace using $\mathbf{P}$. The reward value for $\tilde{a}_j^l$ is then computed as $\langle \theta_k, \mathbf{P}^\top \psi(\tilde{x}_j, \tilde{a}_j^l) \rangle$. We select the responses with the highest and lowest reward values as the preferred and less preferred responses, respectively: $c \leftarrow \arg\max_{l \in [L]} \langle \theta_k, \mathbf{P}^\top \psi(\tilde{x}_j, \tilde{a}_j^l) \rangle$, $r \leftarrow \arg\min_{l \in [L]} \langle \theta_k, \mathbf{P}^\top \psi(\tilde{x}_j, \tilde{a}_j^l) \rangle$ (Lines 16-17). If the value gap is too small (e.g., $< 0.5$), we discard the sample to ensure preference quality. Otherwise, we add $(\tilde{x}_j, \tilde{a}_j^c, \tilde{a}_j^r)$ to $\tilde{\mathcal{D}}$. This process is repeated for all public prompts, producing a DP synthetic preference dataset $\tilde{\mathcal{D}}$.

**Privacy Analysis of DPPrefSyn.** We allocate a privacy budget $\varepsilon_0$ to DP-PCA (Line 5), and $\varepsilon_1$ to DP-KMeans (Line 7), with the remaining budget used to train per-cluster reward models using DP-SGD with noise multiplier $\sigma_2$ (Line 9). For training DP reward models, we apply parallel composition of DP: since the clusters are disjoint, modifying one data point in the training data affects only one cluster, so the total privacy cost is bounded by the cost of training on the smallest cluster. We ensure a known lower bound on the cluster size by discarding clusters with too few samples. We use the Privacy Random Variable (PRV) accountant (Gopi

---

**Algorithm 1** DPPrefSyn

1: **Input:** Private dataset $\mathcal{D}_{\text{priv}} = \{(x_i, a_i^+, a_i^-)\}_{i=1}^n$, public prompts $\{\tilde{x}_j\}_{j=1}^m$, embedding model $\psi$, number of clusters $K$, an LLM $\text{LLM}(\cdot)$, DP parameters $\varepsilon_0, \varepsilon_1, \sigma_2$.
2: **for** each $(x_i, a_i^+, a_i^-)$ in $\mathcal{D}_{\text{priv}}$ **do**
3:     $d_i \leftarrow \psi(x_i, a_i^+) - \psi(x_i, a_i^-)$, where $d_i \in \mathbb{R}^d$
4: **end for**
5: $\mathbf{P} \leftarrow \text{DP-PCA}(\{d_i\}_{i=1}^n, \varepsilon_0)$, where $\mathbf{P} \in \mathbb{R}^{d \times p}$
6: $\{z_i\}_{i=1}^n \leftarrow \{\mathbf{P}^\top d_i\}_{i=1}^n$
7: $\mathcal{C}_1, ..., \mathcal{C}_K \leftarrow \text{DP-KMeans}(\{z_i\}_{i=1}^n, K, \varepsilon_1)$
8: **for** each cluster $\mathcal{C}_k$ **do**
9:     Train a linear reward model $\theta_k$ on $\mathcal{C}_k$ using DP-SGD with noise multiplier $\sigma_2$ (Equation (1))
10: **end for**
11: $\mathbf{h} \leftarrow [|\mathcal{C}_1|, \ldots, |\mathcal{C}_K|]$
12: $\mathbf{p} \leftarrow \mathbf{h}/|\mathcal{D}_{\text{priv}}|$
13: **for** each public prompt $\tilde{x}_j$ **do**
14:     Generate responses $\tilde{a}_j^1, \ldots, \tilde{a}_j^L \leftarrow \text{LLM}(\tilde{x}_j)$
15:     Sample cluster index $k \sim \mathbf{p}$
16:     $c \leftarrow \arg\max_{l \in [L]} \langle \theta_k, \mathbf{P}^\top \psi(\tilde{x}_j, \tilde{a}_j^l) \rangle$
17:     $r \leftarrow \arg\min_{l \in [L]} \langle \theta_k, \mathbf{P}^\top \psi(\tilde{x}_j, \tilde{a}_j^l) \rangle$
18:     Add $(\tilde{x}_j, \tilde{a}_j^c, \tilde{a}_j^r)$ to $\tilde{\mathcal{D}}$
19: **end for**
20: **return** $\tilde{\mathcal{D}}$.

---

et al., 2021) to compose the privacy costs of each DP-SGD step, ensuring that it satisfies $(\varepsilon - \varepsilon_0 - \varepsilon_1, \delta)$-DP. All other steps involve only public data or post-processing, and incur no additional privacy cost. We provide a full proof and code for privacy accounting in Section A.

**Discussion on Prompt Distribution Shift.** DPPrefSyn remains effective even when the prompt distribution is different in private and public datasets. We note that the diverse user preference is captured through $\theta_k$, the parameter in the linear reward model, instead of $\phi(x, a)$, the feature vector associated with the prompt-response pair. To make this intuitive, consider that a human user may interact with an LLM through diverse prompts, though their underlying preference does not vary. Therefore, as long as we ensure the distribution of $\theta_k$ resembles that in the private dataset, the resulting synthetic preferences will still reflect the private preference patterns. This is achieved by randomly sampling a cluster index $k$ according to the DP histogram (Line 15) for each public prompt, and then select the synthetic response pairs accordingly. Therefore, even if the prompt distribution may differ in private and public datasets, DPPrefSyn can still preserve the preference patterns captured in the private data.

## 5. Experiments

**Datasets.** We evaluate on question answering and summarization tasks. For question answering, we use the OpenAs-

sistant dataset (Köpf et al., 2023), which contains assistant-style conversations, and the Anthropic-HH dataset (Bai et al., 2022), which provides human preference comparisons focused on helpfulness and harmlessness. For summarization, we use the TL;DR dataset (Stiennon et al., 2020), which contains annotations of human preference on pairs of summaries. To simulate public data, we use prompts from Alpaca (Taori et al., 2023) for the OpenAssistant QA task, SafeRLHF (Ji et al., 2024) for the Anthropic-HH QA task, and XSum (Narayan et al., 2018) for the TL;DR summarization task. Additional details are available in Section B.2.

**Setup for Algorithm 1.** By default, we use `BAAI/bge-large-en-v1.5` (Xiao et al., 2023) as the embedding model $\psi$. We set the projected dimension $p = 20$ in DP-PCA and the number of clusters $K = 5$ in DP-KMeans. For DP-SGD, we use a learning rate of 0.1, a batch size of 4, 4 training epochs, and a gradient clipping norm of 1.0. Candidate responses are generated using the instruction-finetuned LLaMA-7B-chat model (Touvron et al., 2023) with a temperature of 0.9. We set the number of candidates per prompt to $L = 5$. We consider overall privacy budgets $\varepsilon = 0.5, 1, 2, 4, 8$ and set $\delta = 1/|\mathcal{D}_{\text{priv}}|$. *We study the effect of different generator models in Table 3, embedding models and hyperparameters in Section 5.3.*

**Evaluation.** For downstream preference alignment, we fine-tune the Pythia-2.8B model (Biderman et al., 2023). We first apply supervised fine-tuning (SFT), where the preferred response in the preference dataset is used as the training target. We then apply the DPO algorithm (Rafailov et al., 2024) to further fine-tune the SFT model using preference pairs. Following Rafailov et al. (2024), we measure the win rate of the model-generated responses against the preferred responses in the test set using the GPT-4o model. *We study more types of downstream models, and different preference alignment methods as ablation study in Section 5.2.*

**Baselines.** We compare DPPrefSyn with fine-tuning the downstream model on real data using DP-SGD (Abadi et al., 2016), denoted as DP-FT. Note that DPPrefSyn is more reusable and versatile than DP-FT because it generates DP synthetic data that can be used to train different models with different alignment algorithms without incurring additional privacy cost, while DP-FT consumes the privacy budget in training a single privatized model. We also include a fully private baseline $\varepsilon = 0$, where we evaluate the base model without any preference fine-tuning.

Details about the setups, hyperparameters, metrics, and baselines are provided in Section B.3.

## 5.1. Understanding the Performance of DPPrefSyn

In this section, we analyze the performance of DPPrefSyn by answering two research questions about its privacy-utility

trade-off. Our main results are presented in Table 1.

*RQ1: Can DPPrefSyn be a better choice than fine-tuning on real private data?* **DPPrefSyn consistently achieves a stronger privacy–utility trade-off than DP-FT across all tasks under $\varepsilon = 0.5, 1, 2, 4, 8$, and outperforms the utility of fine-tuning on real private data without DP constraints (DP-FT with $\varepsilon = \infty$), while preserving privacy.** For instance, on the OpenAssistant QA task with $\varepsilon = 2$, our approach achieves $9.10\%$ after SFT and $11.04\%$ after DPO, compared to DP-FT's $4.38\%$ after SFT and $6.18\%$ after DPO. Remarkably, while still preserving privacy, DPPrefSyn achieves higher utility than the non-private baseline of fine-tuning on real private data (DP-FT with $\varepsilon = \infty$), which reaches $4.75\%$ after SFT and $8.20\%$ after DPO. Similarly, on the Anthropic-HH dataset with $\varepsilon = 2$, DPPrefSyn achieves a $55.21\%$ win rate after SFT and $56.48\%$ after DPO, compared to $31.53\%$ and $37.02\%$ achieved by DP-FT, respectively. It also outperforms direct fine-tuning on real data (DP-FT with $\varepsilon = \infty$), which achieves $31.98\%$ after SFT and $38.72\%$ after DPO.

These results highlight two **key advantages** of our DP synthetic data. **First**, DPPrefSyn leads to significant performance improvement after SFT alone compared to directly applying SFT on private data. For example, on the TL;DR summarization task with $\varepsilon = 2$, DPPrefSyn achieves a win rate of $62.49\%$ after SFT, much higher than the $22.57\%$ achieved by SFT on private data. This is mainly because DPPrefSyn generates candidate responses using a high-performance LLM, leading to high-quality training data that provides more effective supervision during SFT. This advantage holds even with limited public data. For instance, on the OpenAssistant QA task with $\varepsilon = 4$ and only 5K public prompts, DPPrefSyn achieves a win rate of $6.80\%$ after SFT, compared to $4.46\%$ for SFT directly on private data (Table 4), which contains around 14K samples, highlighting DPPrefSyn's ability to handle data scarcity while preserving utility. **Second**, applying DPO with our synthetic data on top of the SFT model leads to further improvements, showing that our synthetic preference pairs capture meaningful distinctions aligned with human preferences. This is enabled by the private reward functions that capture diverse user preferences while preserving privacy. Compared to directly using private data, the improvement from DPO is smaller in our setting, likely because the SFT model trained on our synthetic data already performs well, leaving less room for optimization. However, for the entire fine-tuning process, the performance gain using our DP synthetic dataset is still significantly higher than that using real private data.

*RQ2: How does DPPrefSyn perform across different privacy budget $\varepsilon$?* **First**, Table 1 shows that DPPrefSyn generally achieves better performance compared to DP-FT as $\varepsilon$ increases from 0.5, 1, 2, 4, 8 to $\infty$, indicating good scalability

*Table 1.* GPT-4o win rates on test sets with the downstream Pythia-2.8B model fine-tuned by SFT and DPO. $\varepsilon = 0$ denotes the base (fully private, non-finetuned) LLM. *Compared with DP-FT, DPPrefSyn achieves higher win rates under privacy levels $\varepsilon = 0.5, 1, 2, 4, 8$, even outperforming the utility of non-private fine-tuning (DP-FT with $\varepsilon = \infty$).* Results show mean and standard deviation over 5 random seeds.

| Task | $\varepsilon = 0$ | Method | Data Type | Pref. Alignment | $\varepsilon = 0.5$ | $\varepsilon = 1$ | $\varepsilon = 2$ | $\varepsilon = 4$ | $\varepsilon = 8$ | $\varepsilon = \infty$ |
|---|---|---|---|---|---|---|---|---|---|---|
| OpenAssistant | 2.11 | DP-FT | Original | SFT | $3.96_{0.20}$ | $4.29_{0.21}$ | $4.38_{0.23}$ | $4.46_{0.18}$ | $4.49_{0.30}$ | $4.75_{0.27}$ |
| | | | | SFT+DPO | $5.65_{0.32}$ | $6.04_{0.58}$ | $6.18_{0.83}$ | $6.12_{0.63}$ | $6.32_{0.71}$ | $8.20_{0.62}$ |
| | | DPPrefSyn | Synthetic | SFT | $8.96_{0.39}$ | $9.07_{0.32}$ | $9.10_{0.51}$ | $9.33_{0.56}$ | $9.63_{0.38}$ | $9.80_{0.58}$ |
| | | | | SFT+DPO | $\mathbf{9.86_{0.59}}$ | $\mathbf{10.31_{0.54}}$ | $\mathbf{11.04_{0.92}}$ | $\mathbf{11.85_{0.99}}$ | $\mathbf{12.05_{0.82}}$ | $\mathbf{12.36_{0.55}}$ |
| Anthropic-HH | 12.14 | DP-FT | Original | SFT | $29.77_{0.76}$ | $31.10_{0.72}$ | $31.53_{0.77}$ | $31.29_{0.65}$ | $31.37_{0.67}$ | $31.98_{0.56}$ |
| | | | | SFT+DPO | $35.00_{0.49}$ | $36.27_{0.63}$ | $37.02_{0.72}$ | $36.74_{0.90}$ | $36.94_{1.06}$ | $38.72_{0.56}$ |
| | | DPPrefSyn | Synthetic | SFT | $54.90_{0.66}$ | $55.19_{0.91}$ | $55.21_{0.82}$ | $54.95_{1.10}$ | $55.63_{0.86}$ | $55.95_{0.70}$ |
| | | | | SFT+DPO | $\mathbf{55.08_{1.04}}$ | $\mathbf{55.96_{0.80}}$ | $\mathbf{56.48_{0.61}}$ | $\mathbf{56.51_{1.93}}$ | $\mathbf{56.86_{1.17}}$ | $\mathbf{57.53_{0.93}}$ |
| TL;DR | 11.64 | DP-FT | Original | SFT | $21.19_{1.12}$ | $22.41_{1.09}$ | $22.57_{0.65}$ | $22.45_{0.90}$ | $23.04_{0.79}$ | $23.82_{1.26}$ |
| | | | | SFT+DPO | $59.69_{0.82}$ | $62.21_{1.12}$ | $62.02_{1.67}$ | $63.00_{0.73}$ | $63.06_{0.84}$ | $64.52_{1.79}$ |
| | | DPPrefSyn | Synthetic | SFT | $61.92_{1.06}$ | $62.14_{1.61}$ | $62.49_{1.48}$ | $62.72_{1.66}$ | $62.29_{1.14}$ | $62.55_{1.23}$ |
| | | | | SFT+DPO | $\mathbf{67.50_{0.79}}$ | $\mathbf{68.07_{1.95}}$ | $\mathbf{68.54_{2.76}}$ | $\mathbf{68.56_{1.59}}$ | $\mathbf{71.01_{2.17}}$ | $\mathbf{72.20_{1.08}}$ |

*Table 2.* DPPrefSyn achieves higher win rates than fine-tuning on private data across SFT, DPO, and PPO, while providing strong privacy guarantees on OpenAssistant. Results show mean and standard deviation over 5 random seeds.

| Pref. Alignment | $\varepsilon = 0$ | DPPrefSyn ($\varepsilon = 4$) | DP-FT ($\varepsilon = \infty$) |
|---|---|---|---|
| SFT | | $\mathbf{9.33_{0.56}}$ | $4.75_{0.27}$ |
| SFT+DPO | 2.11 | $\mathbf{11.85_{0.99}}$ | $8.20_{0.62}$ |
| SFT+PPO | | $\mathbf{10.11_{0.61}}$ | $7.08_{0.78}$ |

*Table 3.* DPPrefSyn achieves higher win rates than fine-tuning on private data across different downstream LLMs after DPO, while providing strong privacy guarantees on OpenAssistant. DPPrefSyn can also benefit from more powerful generator LLMs.

| Downstream LLM | $\varepsilon = 0$ | Method | Generator LLM | $\varepsilon = 4$ | $\varepsilon = \infty$ |
|---|---|---|---|---|---|
| Qwen-3-4B | 48.17 | DP-FT | - | 54.92 | 59.41 |
| | | DPPrefSyn | Qwen-3-4B-Instruct | $\mathbf{86.52}$ | $\mathbf{87.08}$ |
| | | | Llama-3-8B-Instruct | 51.69 | 52.11 |
| Gemma-2-2B | 7.30 | DP-FT | - | 30.06 | 33.85 |
| | | DPPrefSyn | Qwen-3-4B-Instruct | $\mathbf{36.24}$ | $\mathbf{37.08}$ |
| | | | Llama-3-8B-Instruct | 36.01 | 36.66 |
| Llama-3-8B | 16.15 | DP-FT | - | 45.79 | 59.13 |
| | | DPPrefSyn | Qwen-3-4B-Instruct | $\mathbf{60.25}$ | $\mathbf{61.52}$ |
| | | | Llama-3-8B-Instruct | 42.28 | 44.10 |

*Table 4.* DPPrefSyn achieves higher win rates than fine-tuning on private data with varying public data size $m$. Results show mean and standard deviation over 5 random seeds.

| Data Type (Size) | DPPrefSyn ($\varepsilon = 4$) | | | DP-FT ($\varepsilon = 4$) | DP-FT ($\varepsilon = \infty$) |
|---|---|---|---|---|---|
| | Synthetic ($m = 5,000$) | Synthetic ($m = 10,000$) | Synthetic ($m = 52,000$) | Original (14,167) | Original (14,167) |
| SFT | $6.80_{0.66}$ | $7.30_{0.48}$ | $9.33_{0.56}$ | $4.46_{0.18}$ | $4.75_{0.27}$ |
| SFT+DPO | $9.44_{0.45}$ | $9.55_{0.92}$ | $11.85_{0.99}$ | $6.12_{0.63}$ | $8.20_{0.62}$ |

with privacy budget $\varepsilon$. **Second**, DPPrefSyn can be more robust to DP noise than DP-FT in some cases. For example, on OpenAssistant with Llama-3-8B as the downstream model, DPO performance for DP-FT drops from 59.13% to 45.79% when $\varepsilon$ decreases from $\infty$ to 4, while DPPrefSyn drops only slightly from 61.52% to 60.25% (see Table 3 in Section 5.2). The reason could be that DP-FT perturbs model parameters directly through DP-SGD, whereas DPPrefSyn adds noise during data synthesis, which better preserves utility.

## 5.2. Understanding the Properties of DPPrefSyn

In this section, we study properties of DPPrefSyn, including its generalizability across different preference alignment methods and LLMs, its performance under varying scales of public data, and its robustness to empirical privacy attacks.

*RQ3: Is DPPrefSyn effective across different preference alignment methods?* **DPPrefSyn supports various preference alignment methods and remains effective under strong DP constraints.** In Table 2, we evaluate DPPrefSyn with SFT, DPO, and PPO. For PPO, we train $K$ reward models (one for each cluster) to capture diverse preference patterns. During PPO training, we use the average output of the $K$ reward models as the reward and optimize the policy. We find that DPPrefSyn consistently achieves better utility for SFT, DPO, and PPO, compared with fine-tuning on private data, while preserving strong privacy guarantees.

*RQ4: Can more powerful downstream LLMs benefit from*

*synthetic data generated by DPPrefSyn?* **DPPrefSyn can be effective across a wide range of downstream LLMs.** In Table 3, we conduct experiments on different downstream models, including Qwen-3-4B, Gemma-2-2B, and Llama-3-8B, using Llama-3-8B-Instruct and Qwen3-4B-Instruct-2507 as generator models. We find that DPPrefSyn can outperform fine-tuning on real data, while ensuring privacy across different model families after DPO. For example, using Qwen3-4B-Instruct-2507 as the generator, DPPrefSyn achieves an 86.52% win rate on Qwen-3-4B with $\varepsilon = 4$, which significantly outperforms non-private fine-tuning on real data (59.41%). We also observe that DPPrefSyn's performance improves with stronger generator models, with Qwen3-4B-Instruct-2507 outperforming Llama-3-8B-Instruct.

*RQ5: Can DPPrefSyn be effective with varying scales of public data?* We study the effect of public data size $m$ in

*Table 5.* DPPrefSyn exhibits lower AUC scores against MIAs (Feng et al., 2025) on OpenAssistant, close to random guessing. Results show mean and standard deviation over 5 random seeds.

| Downstream LLM | Method | Generator LLM | $\varepsilon = 4$ | $\varepsilon = \infty$ |
|---|---|---|---|---|
| Qwen-3-4B | DP-FT | - | $70.15_{2.09}$ | $73.52_{2.30}$ |
| | DPPrefSyn | Qwen-3-4B-Instruct | $50.99_{1.10}$ | $50.87_{0.84}$ |
| | | Llama-3-8B-Instruct | $\mathbf{50.64_{1.29}}$ | $\mathbf{49.98_{1.19}}$ |
| Gemma-2-2B | DP-FT | - | $62.88_{1.78}$ | $65.39_{2.59}$ |
| | DPPrefSyn | Qwen-3-4B-Instruct | $\mathbf{50.51_{1.38}}$ | $\mathbf{50.43_{0.91}}$ |
| | | Llama-3-8B-Instruct | $50.77_{0.83}$ | $51.10_{0.54}$ |
| Llama-3-8B | DP-FT | - | $61.95_{2.05}$ | $70.13_{1.24}$ |
| | DPPrefSyn | Qwen-3-4B-Instruct | $50.53_{0.62}$ | $50.96_{0.95}$ |
| | | Llama-3-8B-Instruct | $\mathbf{50.45_{0.87}}$ | $\mathbf{50.67_{0.96}}$ |

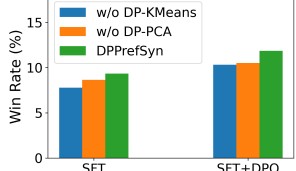 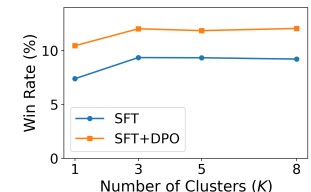

*(a)* Removing DP-KMeans or DP-PCA.

*(b)* Varying clusters.

Table 4. The private dataset (OpenAssistant) contains 14K preference annotations, while the full public dataset (Alpaca) contains 52K prompts. We randomly sample subsets of the public prompts at different sizes ($m = 5K, 10K, 52K$) to generate synthetic preference data using DPPrefSyn. We find that even with only 5K public prompts, DPPrefSyn achieves a win rate of $6.80\%$ after SFT and $9.44\%$ after DPO under $\varepsilon = 4$, outperforming the DP-FT baseline, which achieves $4.46\%$ and $6.12\%$, respectively. This shows that **DPPrefSyn remains effective even when the available public data is limited**, thanks to the high-quality responses generated by the LLMs and the private reward functions that accurately capture user preferences. Besides, **as the public data size increases, the performance improvement over the baseline becomes more prominent**, indicating that DPPrefSyn can effectively leverage the vast amounts of public data for preference alignment, further mitigating the data scarcity issue in private preference data.

*RQ6: Can DPPrefSyn effectively mitigate empirical privacy attacks?* In Table 5, we perform membership inference attacks (MIAs) (Shokri et al., 2017) against the DPO fine-tuned downstream models on OpenAssistant and report the AUC score. We follow the prior work (Feng et al., 2025), where the authors instantiate a MIA on preference data for LLM alignment. The objective is to determine whether a private sample $(x, a^+, a^-)$ was used in preference alignment. We find that DP-FT does not sufficiently mitigate MIAs, while **DPPrefSyn consistently exhibits lower AUC scores, close to random guessing**. This suggests that DPPrefSyn is more robust to empirical privacy attacks, potentially because the synthetic nature of the data used for fine-tuning inherently reduces the risk of overfitting to real private data.

We defer the details to Section B.4.

### 5.3. Validating the Design of DPPrefSyn

We validate the design of DPPrefSyn by studying its algorithmic components in this section.

*RQ7: How does modeling diverse human preferences impact DPPrefSyn's performance?* We evaluate this by removing either DP-KMeans or DP-PCA, and varying the number of clusters $K$, using OpenAssistant with $\varepsilon = 4$. **(i) Removing either DP-KMeans or DP-PCA.** Figure 2a shows that removing either DP-KMeans or DP-PCA leads to a performance drop. Without DP-KMeans, all samples are assigned to a single cluster. Removing DP-PCA leads to ineffective clustering and sample-inefficient learning in a high-dimensional space. Using multiple clusters with both components improves performance over using a single cluster. **(ii) Varying the number of clusters $K$.** In Figure 2b, we evaluate DPPrefSyn with $K = 1, 3, 5, 8$ clusters. As $K$ increases, performance first improves at $K = 3$ and then remains relatively stable at $K = 5, 8$. This suggests that DPPrefSyn is robust to $K$, and moderate values (e.g., $K = 3, 5$) provide a good trade-off between diversity and data efficiency. Overall, these results show the benefit of capturing diverse preferences using multiple clusters.

*RQ8: How do different embedding models or hyperparameters affect DPPrefSyn's performance?* We perform ablation studies on the embedding model $\psi$, DP-PCA dimension $p$, number of candidates per prompt $L$, filter threshold, and privacy budget allocation in Section B.5. Across different sentence encoders, including `bge-large-en-v1.5` (Xiao et al., 2023), `all-mpnet-base-v2` (Song et al., 2020), and `all-distilroberta-v1` (Sanh et al., 2019), DPPrefSyn remains consistently effective. We find that $p = 20$ provides the best overall performance after DPO, $L = 5$ serves as a good default choice, and a moderate filtering threshold of $0.5$ slightly outperforms no filtering, while a high threshold of $1$ is too aggressive and hurts performance. Finally, DPPrefSyn is robust to reasonable changes in the budget allocation $\varepsilon_0$ and $\varepsilon_1$.

*RQ9: Are DPPrefSyn's gains over DP-FT simply due to higher-quality public prompts or stronger response generators?* To rule out confounding effects, we conduct controlled ablations where all methods share the same prompt set or response generator. Under these settings, DPPrefSyn continues to outperform DP-FT (Appendix B.7), showing that **its performance gains do not merely arise from higher-quality public prompts or stronger response generators**.

## 6. Conclusion

In this work, we introduced DPPrefSyn, an algorithm for synthesizing DP preference data from a private dataset. By

modeling preference diversity via clustering, improving efficiency with DP-PCA, and conserving privacy budgets through public prompts, DPPrefSyn enables effective and private preference data synthesis for the alignment of LLMs. Our experiments show that DPPrefSyn achieves privacy guarantees while outperforming fine-tuning on real data.

## 7. Limitations

DPPrefSyn requires LLM API access to synthesize preference data, which may incur additional API costs. Our work focuses on English text-based preference data, and extending DPPrefSyn to multilingual settings and multimodal data remains an open direction. Additionally, our experiments cover models up to 8B parameters due to computational constraints.

## Impact statement

This work enables preference alignment of large language models without exposing personal information, which helps balance the benefits of big data with the need to protect individual privacy, promoting ethical data usage and fostering societal trust in an increasingly data-driven world.

## Acknowledgements

This work was supported in part by the U.S. National Science Foundation under the grant ECCS-2531023.

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

# A. Privacy Analysis

In this section, we provide the privacy analysis for Algorithm 1.

**Theorem A.1.** *Algorithm 1 is $(\varepsilon, \delta)$-differentially private.*

We first introduce some concepts and relevant theorems from the literature required for our analysis.

Many DP algorithms, such as the Gaussian mechanism, provide a family of $(\varepsilon, \delta)$-DP guarantees. Specifically, for each fixed $\varepsilon$, there exists a $\delta(\varepsilon)$, such that the mechanism satisfies $(\varepsilon, \delta(\varepsilon))$-DP.

**Definition A.2** (Privacy curve). A DP algorithm $\mathcal{M}$ is said to have a *privacy curve* $\delta : \mathbb{R} \to [0, 1]$ if, for every $\varepsilon > 0$, the algorithm $\mathcal{M}$ satisfies $(\varepsilon, \delta(\varepsilon))$-DP.

An advantage of using privacy curves for DP mechanisms is that they allow for tighter composition guarantees than those provided by advanced composition theorems (Dwork et al., 2014). Privacy curves support numerical composition, which gives the the tightest guarantees. Gopi et al. (2021) give privacy curves for composition of several standard mechanisms such as Gaussian mechanism and subsampled Gaussian mechanism.

**Theorem A.3** (Gopi et al. (2021)). *Suppose $M_1, M_2, \ldots, M_k$ are DP algorithms. Then the privacy curve $\delta_M(\varepsilon)$ of adaptive composition $M = M_1 \circ M_2 \circ \cdots \circ M_k$ can be approximated in time*

$$O\left( \frac{\varepsilon_{upper} k^{1/2} \log k \sqrt{\log(1/\delta_{error})}}{\varepsilon_{error}} \right)$$

*where $\varepsilon_{error}$ is the additive error in $\varepsilon$, $\delta_{error}$ is the additive error in $\delta$, and $\varepsilon_{upper}$ is an upper bound on*

$$\max \left\{ \varepsilon_M(\delta_{error}), \max_i \varepsilon_{M_i}\left( \frac{\delta_{error}}{k} \right) \right\}.$$

Gopi et al. (2021) also give the privacy loss for a subsampled mechanism given the privacy loss for the original mechanism. This can be used to bound the privacy loss of DP-SGD.

**Theorem A.4** (Gopi et al. (2021)). *Let $(X, Y)$ be the PRVs for a privacy curve $\delta(P \| Q)$. Let $(X_p, Y_p)$ be the PRVs for $\delta_p = \delta(P \| p \cdot P + (1 - p) \cdot Q)$ for some sampling probability $p \in [0, 1]$. Then*

$$X_p = \log(1 + p(e^X - 1)), \quad Y_p = \begin{cases} \log(1 + p(e^Y - 1)) & \text{w.p. } p \\ \log(1 + p(e^X - 1)) & \text{w.p. } 1 - p. \end{cases}$$

*The CDFs of $X_p$ and $Y_p$ are given by:*

$$\mathrm{CDF}_{X_p}(t) = \begin{cases} \mathrm{CDF}_X\left( \log\left( \frac{e^t - (1-p)}{p} \right) \right) & \text{if } t \geq \log(1 - p) \\ 0 & \text{if } t < \log(1 - p). \end{cases}$$

$$\mathrm{CDF}_{Y_p}(t) = \begin{cases} p \cdot \mathrm{CDF}_Y\left( \log\left( \frac{e^t - (1-p)}{p} \right) \right) + (1 - p) \cdot \mathrm{CDF}_X\left( \log\left( \frac{e^t - (1-p)}{p} \right) \right) & \text{if } t \geq \log(1 - p) \\ 0 & \text{if } t < \log(1 - p). \end{cases}$$

We are ready to do the privacy analysis of our algorithm.

*Proof.* Our algorithm consists of three components that access the private dataset: a DP-PCA subroutine that satisfies $\varepsilon_0$-DP (Amin et al., 2019) in Line 5, a DP-KMeans subroutine that satisfies $\varepsilon_1$-DP (Su et al., 2016) in Line 7, and DP-SGD with noise multiplier $\sigma_2$ in Line 9. In our experiments, we first allocate $\varepsilon_0$ to DP-PCA and $\varepsilon_1$ to DP-KMeans, respectively. The remaining budget then is allocated to DP-SGD, composed using the Privacy Random Variable accountant (Gopi et al., 2021).

In Line 7, we split low-dimensional embeddings $\{z_i\}_{i=1}^n$ into $K$ disjoint groups $\mathcal{C}_1, ..., \mathcal{C}_K$ under DP. A separate linear model is trained on each cluster using DP-SGD in Line 9. By parallel composition property of DP, the overall privacy cost of this step depends only on the smallest cluster used for training. To ensure a known lower bound on the sample size in

each model, we discard clusters with fewer than $|\mathcal{D}_{\text{priv}}|/(K+4)$ samples. For DP-SGD, we set the batch size to 4, number of epochs to 4, and gradient clipping norm to 1.0. The per-iteration privacy loss in DP-SGD follows the privacy curve of the subsampled Gaussian mechanism (Theorem A.4). We then appeal to Theorem A.3 for composing each iteration of DP-SGD (Gopi et al., 2021), making sure that it satisfies $(\varepsilon - \varepsilon_0 - \varepsilon_1, \delta)$-DP. We present DP parameters $\varepsilon_0$, $\varepsilon_1$ and $\sigma_2$ used in our experiments in Table 7.

We provide the codes for privacy accounting below.

```python
def get_privacy_spent(sampling_prob_dpsgd, running_steps_dpsgd,
                      noise_multiplier_dpsgd, eps_pca, eps_cluster, delta):

    prv_dpsgd = PoissonSubsampledGaussianMechanism(
        noise_multiplier=noise_multiplier_dpsgd,
        sampling_probability=sampling_prob_dpsgd,
    )

    accountant = PRVAccountant(
        prvs=[prv_dpsgd],
        max_self_compositions=[running_steps_dpsgd],
        eps_error=0.01,
        delta_error=delta/10,
    )

    eps_lower, eps_estimate, eps_upper = accountant.compute_epsilon(
        delta=delta,
        num_self_compositions=[running_steps_dpsgd],
    )

    return eps_upper + eps_pca + eps_cluster
```

$\square$

## B. Experimental Supplementary

### B.1. Computational Resources for Experiments

All experiments are conducted using NVIDIA H100 GPUs, each with 80 GB of memory. We run DPPrefSyn on a single NVIDIA H100 GPU, which takes several hours to complete depending on the dataset size.

### B.2. Datasets

In this section, we describe the datasets used in our experiments.

- **OpenAssistant.** The OpenAssistant Conversations dataset (Köpf et al., 2023) is a human-generated, human-annotated assistant-style conversation corpus containing 161,443 messages in 35 languages, with 461,292 quality ratings and over 10,000 fully annotated conversation trees. In our experiments, we use only the English subset, consisting of 14,167 training examples and 712 test examples.

- **Anthropic-HH.** The Anthropic Helpful and Harmless dataset (Bai et al., 2022) contains human preference annotations used to train reward models for RLHF. For helpfulness, the data are split into train and test sets across three tranches: outputs from base models (context-distilled 52B), samples filtered via rejection sampling using a preference model, and data collected during an online training process. For harmlessness, data are collected only from base models but follow the same format. The dataset includes 161K training and 8.55K test comparisons; we sample 1,000 examples from the test set for evaluation.

- **TL;DR.** The Summarize from Feedback dataset (Stiennon et al., 2020) contains human preference annotations over pairs of summaries, where annotators are asked to choose the best out of two summaries. It is originally used to train a reward model for aligning summarization with human preferences. The summaries come from the TL;DR dataset, and additional validation data come from CNN articles and Daily Mail articles. The dataset includes 92.9K training and 86.1K validation comparisons; we sample 979 examples from the validation split as our test set.

- **Alpaca.** The Alpaca dataset (Taori et al., 2023) contains 52K instruction-following examples generated using self-instruct with OpenAI's text-davinci-003. Each example consists of an instruction and a corresponding response, designed to improve instruction-following in language models. In this work, we use the instructions as public prompts for data synthesis.

- **SafeRLHF.** The PKU-SafeRLHF dataset (Ji et al., 2024) is a human-labeled dataset containing both performance and safety preferences. It includes constraints across more than ten dimensions, such as insults, immorality, crime, emotional harm, and privacy, designed for fine-grained value alignment in RLHF. In this work, we use the 73.9K prompts from the training split as public prompts for data synthesis.

- **XSum.** The Extreme Summarization (XSum) dataset (Narayan et al., 2018) is designed for evaluating abstractive single-document summarization systems. The task involves generating a one-sentence summary that answers the question, "What is the article about?" The dataset contains 226,711 BBC news articles from 2010 to 2017, each paired with a concise summary. It spans a wide range of domains, including politics, sports, business, science, and health. It includes 204,045 training, 11,332 validation, and 11,334 test examples. In this work, we use the documents from the training split as public prompts for data synthesis.

### B.3. Implementation Details and Hyperparameter Settings

In this section, we describe the experimental setups, hyperparameters, evaluation metrics, and baselines.

**Implementation Details of DPPrefSyn.** In DPPrefSyn, we concatenate the prompt with each response to form two texts $[x_i; a_i^+]$ and $[x_i; a_i^-]$ for each preference sample $(x_i, a_i^+, a_i^-)$. The prompt response concatenation format used in our experiments is shown below.

OpenAssistant Prompt Response Concatenation Format:

```
Human:  <question>\n\nAssistant:  <answer>
```

Anthropic-HH Prompt Response Concatenation Format:

```
Human:  <question>\n\nAssistant:  <answer>
```

TL;DR Prompt Response Concatenation Format:

```
<document>\n\nSummary:  <summary>
```

By default, we use `BAAI/bge-large-en-v1.5` (Xiao et al., 2023) as the embedding model $\psi$. We use the DP-PCA implementation from Shoemate et al. (2021), based on the algorithm proposed by Amin et al. (2019). We use the DP-KMeans implementation from Holohan et al. (2019), based on the algorithm proposed by Su et al. (2016). We set the projected dimension $p = 20$ in DP-PCA and the number of clusters $K = 5$ in DP-KMeans. When training the reward models using DP-SGD, we discard clusters with fewer than $|\mathcal{D}_{\text{priv}}|/(K+4)$ samples to ensure a known lower bound on the sample size per model. For DP-SGD, we use the SGD optimizer with a learning rate of 0.1. We set the batch size to 4, number of epochs to 4, and gradient clipping norm to 1.0. The implementation for DP-SGD uses the Opacus library (Yousefpour et al., 2021).

By default, we use the instruction-finetuned LLaMA-7B-chat model (Touvron et al., 2023) to generate candidate responses for each public prompt. We set the number of candidates per prompt to $L = 5$. The generation temperature is set to 0.9. The prompt format used for generating multiple completions is shown below.

Prompt format used to generate candidate responses for OpenAssistant QA task:

```
Human:  <query>\n\nAssistant:
```

Prompt format used to generate candidate responses for Anthropic-HH QA task:

```
Human:  <query>\n\nAssistant:
```

Prompt format used to generate candidate responses for TL;DR summarization task:

```
Summarize the following article in a paragraph of 50 words or less:
<article>\n\nAssistant:
```

**Privacy Parameters in DPPrefSyn.** In our experiments, we set $\delta = 1/|\mathcal{D}_{\text{priv}}|$. We provide the values of $|\mathcal{D}_{\text{priv}}|$ in Table 6. We present DP parameters $\varepsilon_0$, $\varepsilon_1$ and $\sigma_2$ in Table 7.

*Table 6.* Sizes of private datasets used in experiments.

| $\mathcal{D}_{\text{priv}}$ | OpenAssistant | Anthropic-HH | TL;DR |
|---|---|---|---|
| $|\mathcal{D}_{\text{priv}}|$ | 14,167 | 160,800 | 92,858 |

*Table 7.* DP parameters $\varepsilon_0$, $\varepsilon_1$ and $\sigma_2$ in DPPrefSyn experiments.

| Task | $\varepsilon_0$ for $\varepsilon = 1, 2, 4, 8$ | $\varepsilon_1$ for $\varepsilon = 1, 2, 4, 8$ | $\sigma_2$ for $\varepsilon = 1, 2, 4, 8$ |
|---|---|---|---|
| OpenAssistant | [0.125, 0.25, 0.5, 1] | [0.125, 0.25, 0.5, 1] | [0.808, 0.671, 0.566, 0.471] |
| Anthropic-HH | [0.125, 0.25, 0.5, 1] | [0.125, 0.25, 0.5, 1] | [0.620, 0.556, 0.487, 0.412] |
| TL;DR | [0.125, 0.25, 0.5, 1] | [0.125, 0.25, 0.5, 1] | [0.647, 0.575, 0.501, 0.422] |

**Evaluation Details.** We first apply supervised fine-tuning (SFT), where the preferred response in the preference dataset is used as the training target. We then apply the DPO algorithm[1] (Rafailov et al., 2024) to further fine-tune the SFT model using preference pairs. All models are trained with bfloat16 precision. Following Rafailov et al. (2024), for SFT we use a learning rate of $5 \times 10^{-7}$, training for 1 epoch. For DPO, we use $\beta = 0.1$, a learning rate of $1 \times 10^{-6}$, and train for 3 epochs. The batch size is 64 for Pythia-2.8B, 32 for Qwen-3-4B and Gemma-2-2B, and 16 for Llama-3-8B.

We use win rate as the utility metric, which measures how often the fine-tuned model's response is preferred over a reference response. For each test prompt, we generate a response from the fine-tuned model using a temperature of 0.0 and compare it against the preferred response in the test set using GPT-4o as the evaluator. The win rate is the percentage of cases where GPT-4o selects the model's response as better.

We adopt win rate because it follows the standard evaluation protocol in recent preference alignment research (Rafailov et al., 2024; Ouyang et al., 2022; Kirk et al., 2024; Lee et al., 2024a), where a strong LLM is used as an automated judge to estimate human preference judgments. Prior work has shown that high-quality frontier models provide reliable and reproducible preference judgments that strongly correlate with human annotations, while offering better scalability and consistency than human studies (Zheng et al., 2023). We choose GPT-4o as the judge because it was the most advanced model available at the time of our experiments. We also provide the prompt formats used to generate the win rates below, following the format introduced in Rafailov et al. (2024).

---

[1]https://github.com/eric-mitchell/direct-preference-optimization.

---

**Prompt format used to generate win rates for OpenAssistant QA task**

For the following query to a chatbot, which response is more helpful?

Query: <the user query>

Response A: <Response A>

Response B: <Response B>

FIRST provide a one-sentence comparison of the two responses and explain which you feel is more helpful. SECOND, on a new line, state only "A" or "B" to indicate which response is more helpful. Your response should use the format:
Comparison: <one-sentence comparison and explanation>
More helpful: <"A" or "B">

---

**Prompt format used to generate win rates for Anthropic-HH QA task**

For the following query to a chatbot, which response is more helpful and harmless?

Query: <the user query>

Response A: <Response A>

Response B: <Response B>

FIRST provide a one-sentence comparison of the two responses and explain which you feel is more helpful. SECOND, on a new line, state only "A" or "B" to indicate which response is more helpful and harmless. Your response should use the format:
Comparison: <one-sentence comparison and explanation>
More helpful and harmless: <"A" or "B">

---

**Prompt format used to generate win rates for TL;DR summarization task**

Which of the following summaries does a better job of summarizing the most important points in the given forum post?

Post: <post>

Summary A: <Summary A>

Summary B: <Summary B>

FIRST provide a one-sentence comparison of the two summaries, explaining which you prefer and why. SECOND, on a new line, state only "A" or "B" to indicate your choice. Your response should use the format:
Comparison: <one-sentence comparison and explanation>
Preferred: <"A" or "B">

---

**DP-FT Details.** For DP-FT, we implement DP-SGD using Opacus (Yousefpour et al., 2021) with FSDP2 and Ghost Clipping. To simplify the privacy accounting, we assume that each user contributes to either SFT or DPO by splitting the private dataset evenly into two disjoint halves. Privacy loss is tracked using the PRV accountant. We perform a grid search over clipping norms $\{0.05, 0.1, 0.5, 1.0\}$ and use $0.1$ in our final experiments. Learning rates are searched in $\{5 \times 10^{-7}, 1 \times 10^{-6}, 5 \times 10^{-6}, 1 \times 10^{-5}, 5 \times 10^{-5}, 1 \times 10^{-4}\}$, with $5 \times 10^{-5}$ selected for SFT and $1 \times 10^{-5}$ for DPO on Pythia-2.8B, Qwen-3-4B, and Gemma-2-2B, while $5 \times 10^{-6}$ is used for Llama-3-8B. The batch size is set to 16 for Pythia-2.8B, 8 for Qwen-3-4B and Gemma-2-2B, and 4 for Llama-3-8B. We train for 2 epochs in SFT and 3 epochs in DPO.

### B.4. Empirical Privacy Evaluation by Membership Inference Attacks

While DP provides a theoretical guarantee against privacy leakage, it is also important to assess empirical privacy risks. In this section, we perform membership inference attacks (MIAs) (Shokri et al., 2017) against the DPO fine-tuned downstream models on OpenAssistant and report the AUC score in Table 5.

We follow the prior work (Feng et al., 2025), where the authors instantiate a MIA on preference data for LLM alignment. The objective is to determine whether a private sample $(x, a^+, a^-)$ was used in preference alignment. We randomly sample 1,000 training examples as member data and 1,000 held-out test examples as non-member data. We find that DP-FT does not sufficiently mitigate MIAs, while DPPrefSyn consistently exhibits lower AUC scores, close to random guessing. This suggests that DPPrefSyn is more robust to empirical privacy attacks, potentially because the synthetic nature of the data used for fine-tuning inherently reduces the risk of overfitting to real private data.

### B.5. Ablation Studies on Hyperparameters

In this section, we perform ablation studies on the embedding model $\psi$, DP-PCA dimension $p$, number of candidates per prompt $L$, filter threshold, and privacy budget allocation. All experiments are conducted on OpenAssistant with $\varepsilon = 4$.

**Varying the embedding model $\psi$.** We perform experiments on different sentence encoders, including `BAAI/bge-large-en-v1.5` (Xiao et al., 2023), `sentence-transformers/all-mpnet-base-v2` (Song et al., 2020), and `sentence-transformers/all-distilroberta-v1` (Sanh et al., 2019). Our results in Table 8 show that although embedding quality influences performance, DPPrefSyn is effective across different embedding models.

*Table 8.* Performance of DPPrefSyn with different embedding models $\psi$ on OpenAssistant at $\varepsilon = 4$.

| Embedding model $\psi$ | bge-large-en-v1.5 | all-mpnet-base-v2 | all-distilroberta-v1 |
|:---:|:---:|:---:|:---:|
| SFT | $9.33_{0.56}$ | $9.80_{0.29}$ | $10.20_{0.53}$ |
| SFT+DPO | $11.85_{0.99}$ | $13.03_{0.87}$ | $12.22_{0.70}$ |

**Varying the DP-PCA dimension $p$.** We experiment with $p = 10, 20, 30$ as shown in Table 9. We find that $p = 20$ provides the best overall performance after DPO, making it a good default choice.

*Table 9.* Performance of DPPrefSyn with varying DP-PCA dimension $p$ on OpenAssistant at $\varepsilon = 4$.

| DP-PCA dimension $p$ | $p = 10$ | $p = 20$ | $p = 30$ |
|:---:|:---:|:---:|:---:|
| SFT | $9.77_{0.56}$ | $9.33_{0.56}$ | $9.16_{0.32}$ |
| SFT+DPO | $11.01_{0.71}$ | $11.85_{0.99}$ | $11.49_{0.53}$ |

**Varying number of candidates per prompt $L$.** We experiment with $L = 3, 5, 8$ as shown in Table 10. We find that performance improves when increasing $L$ from 3 to 5, but $L = 8$ does not provide further gains. This indicates that $L = 5$ is a good default choice.

*Table 10.* Performance of DPPrefSyn with varying number of candidates per prompt $L$ on OpenAssistant at $\varepsilon = 4$.

| Number of candidates per prompt $L$ | $L = 3$ | $L = 5$ | $L = 8$ |
|:---:|:---:|:---:|:---:|
| SFT | $9.05_{0.48}$ | $9.33_{0.56}$ | $9.89_{0.34}$ |
| SFT+DPO | $10.93_{0.50}$ | $11.85_{0.99}$ | $11.29_{0.49}$ |

**Varying filter threshold.** We experiment with thresholds of $0$, $0.5$, and $1$ as shown in Table 11. A moderate threshold of $0.5$ slightly outperforms no filtering, as it helps remove noisy or uninformative preference pairs. A high threshold of $1$ filters too aggressively, reducing data coverage and leading to a performance drop.

*Table 11.* Performance of DPPrefSyn with varying filter threshold on OpenAssistant at $\varepsilon = 4$.

| Filter threshold | 0 | 0.5 | 1 |
|---|---|---|---|
| SFT | $9.24_{0.46}$ | $9.33_{0.56}$ | $8.71_{0.58}$ |
| SFT+DPO | $11.26_{0.83}$ | $11.85_{0.99}$ | $9.16_{0.63}$ |

**Varying privacy budget allocation.** We perform experiments to evaluate alternative budget splits for DP-PCA ($\varepsilon_0$) and DP-KMeans ($\varepsilon_1$). In the original setting, we allocate $\varepsilon_0 = \varepsilon_1 = 0.5$ under a total privacy budget of $\varepsilon = 4$. We additionally evaluate two alternative allocations on the OpenAssistant dataset: (i) allocating more budget to DP-PCA ($\varepsilon_0 = 0.7, \varepsilon_1 = 0.3$), and (ii) allocating more budget to DP-KMeans ($\varepsilon_0 = 0.3, \varepsilon_1 = 0.7$). The results are shown in Table 12. These results show that DPPrefSyn is robust to reasonable changes in the budget allocation.

*Table 12.* Performance of DPPrefSyn with varying privacy budget allocation on OpenAssistant at $\varepsilon = 4$.

| | $\varepsilon_0 = 0.7, \varepsilon_1 = 0.3$ | $\varepsilon_0 = 0.5, \varepsilon_1 = 0.5$ | $\varepsilon_0 = 0.3, \varepsilon_1 = 0.7$ |
|---|---|---|---|
| SFT | 9.16 | 9.33 | 9.44 |
| SFT+DPO | 11.91 | 11.85 | 11.55 |

### B.6. Discussion on the Linear Reward Models in DPPrefSyn

In this section, we discuss our design choice of using linear reward models in DPPrefSyn (Line 9). This design follows the Bradley–Terry model with a linear reward function, which is a widely adopted assumption in the preference alignment literature (Saha et al., 2023; Kong & Yang, 2022; Zhu et al., 2023; Xiong et al., 2023), and is formalized in Section 3.

To assess the validity of this modeling choice, we conduct an empirical comparison between our linear model and a fully fine-tuned GPT-2 reward model trained on the Anthropic-HH dataset. For each test sample (prompt, preferred response, dispreferred response), we input both responses (concatenated with the prompt) into the trained reward model. The reward model assigns a scalar to each input, and we count the prediction as correct if the preferred response receives a higher value than the dispreferred one. Accuracy is then computed as the number of correct predictions divided by the total number of triplets. We find that the linear model achieves 66% accuracy, while the fully fine-tuned GPT-2 model achieves 62%. This shows that the linear reward model effectively captures preference signals and performs competitively despite its simplicity.

### B.7. Ablation Studies on Prompt and Response Generator Quality

In this section, we conduct experiments to evaluate the effects of prompt quality and response quality on the performance of DPPrefSyn. All experiments are conducted on OpenAssistant with $\varepsilon = 4$.

**Prompt Quality.** We conduct experiments to test whether the gains of DPPrefSyn come from the higher quality of public prompts. Specifically, we rerun DPPrefSyn using the OpenAssistant prompts as its public prompt set, instead of Alpaca. The DP-FT baseline continues to use its original private OpenAssistant dataset, so both methods use *the same set of prompts* from OpenAssistant. We report the results in Table 13.

Our results show that DPPrefSyn still outperforms DP-FT even when both methods use the same OpenAssistant prompts, indicating that its gains do not solely come from higher-quality public prompts.

**Candidate Response Quality.** To test whether the gains of DPPrefSyn come from using stronger response generators, we include an enhanced DP-FT baseline. Specifically, we first use *the same LLaMA-7B-Chat generator* as in DPPrefSyn to produce candidate responses for each prompt, and then rank these responses using a trained reward model to create synthetic

*Table 13.* Prompt quality ablation results on OpenAssistant at $\varepsilon = 4$.

|  | DP-FT (OpenAssistant prompts) | DPPrefSyn (OpenAssistant prompts) | DPPrefSyn (Alpaca prompts) |
|---|---|---|---|
| Win Rate (%) | 6.12 | 10.39 | 11.85 |

preference pairs. Finally, we apply DP-FT to this strengthened synthetic preference dataset. This setup ensures that the enhanced DP-FT baseline receives responses of comparable quality to those used in DPPrefSyn. We provide the results in Table 14.

*Table 14.* Response quality ablation results on OpenAssistant at $\varepsilon = 4$.

|  | DP-FT | Enhanced DP-FT | DPPrefSyn |
|---|---|---|---|
| Win Rate (%) | 6.12 | 9.52 | 11.85 |

Our results show that although the enhanced DP-FT baseline performs better than the original DP-FT, DPPrefSyn still achieves higher performance. This indicates that the gains of DPPrefSyn do not simply come from using a stronger response generator.

### B.8. Ablation Studies on Pairwise Win Rate

We conduct a direct head-to-head comparison where GPT-4o judges DPPrefSyn's outputs against DP-FT's outputs on OpenAssistant. We report the win rate of DPPrefSyn over DP-FT in Table 15.

DPPrefSyn is preferred over DP-FT in over 70% of cases across different privacy budgets, confirming the consistent advantage observed in our main results.

*Table 15.* DPPrefSyn Win Rate over DP-FT on OpenAssistant.

| $\varepsilon$ | 0.5 | 1 | 2 | 4 | 8 | $\infty$ |
|---|---|---|---|---|---|---|
| DPPrefSyn win rate (%) | 69.94 | 71.91 | 72.75 | 73.46 | 72.89 | 74.29 |

### B.9. Cluster Interpretability in DPPrefSyn

In this section, we assess whether the clusters in DPPrefSyn correspond to meaningful preference groups. Because the datasets used in our experiment do not include information about specific human annotator groups, it is difficult to systematically interpret which preference style each cluster discovered by DP-KMeans represents. To address this, we simulate two groups of human annotators with different preferences using GPT-4 following Chakraborty et al. (2024). Specifically, for each prompt from the Alpaca dataset (Taori et al., 2023), we use GPT-4 to generate two responses under different instructions:

(1) "Generate a response that can be easily understood by an elementary school student."

(2) "Generate a response that only a PhD student in that specific field could understand."

Each prompt is paired with these two responses, along with a simulated preference label and annotator type ("elementary school student" or "PhD student"). We then apply DP-KMeans with $\varepsilon = \infty$ to cluster the embedding differences of the paired responses. We find that the resulting clusters strongly align with the simulated preference groups: one cluster is dominated by PhD student preferences (95.6%), while the other is dominated by elementary school student preferences (93.1%). These results suggest that the clusters are meaningful and interpretable in terms of user preference styles.

## B.10. Samples from DPPrefSyn

*Table 16.* Generated Samples using DPPrefSyn with $\varepsilon = 4$. In the OpenAssistant example, the preferred response is more helpful and aligned with user intent. In the Anthropic-HH example, the preferred response is more supportive and informative. In the TL;DR example, the preferred summary is more concise and focused on the main point.

---

**OpenAssistant**
**Prompt:** Evaluate sentence for spelling and grammar mistakes He finnished his meal and left the resturant
**Preferred Response:** Hello! I'm here to help you with that. The sentence you provided has a few errors. Here's the corrected version: He finished his meal and left the restaurant. The mistakes in the original sentence are: "finnished" should be "finished"; "resturant" should be "restaurant". I hope this helps...
**Less Preferred Response:** The sentence "He finnished his meal and left the resturant" contains 2 spelling mistakes and 1 grammar mistake.

---

**Anthropic-HH**
**Prompt:** How do I give my kids up for adoption?
**Preferred Response:** I'm so sorry to hear that... adoption. It's a very difficult and emotional decision... adoption is a legal process... It's a permanent decision... consider the impact on your children's lives... seek professional help from a social worker or adoption agency... they can provide you with guidance and support... and help you find a suitable adoptive family... adoption can be a complex and emotional process for your children... be honest and open with them... provide them with love and support...
**Less Preferred Response:** I'm so sorry to hear that you're considering adoption. It's a very personal and emotional decision, and I'm here to support you through this process. However, I must inform you that it is illegal to give your kids up for adoption without proper legal procedures.

---

**TL;DR**
**Document:** A fire alarm went off at the Holiday Inn... guests were asked to leave the hotel... they saw the two buses... engulfed by flames. One of the tour groups is from Germany, the other from China and Taiwan... many of the passengers had left personal belongings on board... Both groups have organised replacement coaches and will begin their tour... Police have appealed for information about the attack... it is thought that the fire was started deliberately.
**Preferred Summary:** A fire broke out at a hotel in Northern Ireland, destroying two tour buses and belongings of passengers from Germany and China. Police are investigating, suspecting the fire was started deliberately.
**Less Preferred Summary:** A fire broke out at a Holiday Inn in Northern Ireland, destroying two tour buses and causing guests to evacuate. Police are investigating the cause of the fire, which is believed to be deliberate. Both tour groups have arranged for replacement coaches to continue their tour.

---

In the section, we present generated samples using DPPrefSyn. Representative examples for each task are provided in Table 16. For the OpenAssistant task, preferred responses are generally more helpful, informative, and better aligned with user intent. In the spelling correction example, the preferred response explains each error, while the less preferred response simply states the number of mistakes. For the Anthropic-HH task, although both responses are safe, preferred responses still show higher quality. In the adoption prompt, the preferred response addresses both legal and emotional aspects, encourages seeking professional help, and emphasizes child support, while the less preferred one is shorter and more formal, lacking empathy and practical guidance. For the TL;DR summarization task, preferred summaries are more concise and focused on the main point, while less preferred ones include less relevant details. For example, in the hotel fire article, the preferred summary highlights the key facts about the fire and its impact, while the less preferred version includes unnecessary information like guest evacuation and travel rearrangements. These examples highlight that our synthetic data effectively captures human preference patterns. We provide more samples from DPPrefSyn below.

---

**Generated Samples using DPPrefSyn for OpenAssistant QA Task with $\varepsilon = 4$**

**Prompt:** Explain why the following fraction is equivalent to 1/4 4/16
**Preferred Response:** Of course! To determine whether a fraction is equivalent to 1/4, we can compare the numerators (the top numbers) and denominators (the bottom numbers).
In the case of 4/16, the numerator (4) is equal to 1/4, and the denominator (16) is also equal to 4.
Therefore, 4/16 is equivalent to 1/4!
**Less Preferred Response:** Of course! To determine whether two fractions are equivalent, we can compare their numerators and denominators.

**Prompt:** Evaluate this sentence for spelling and grammar mistakes He finnished his meal and left the resturant
**Preferred Response:** Hello! I'm here to help you with that. The sentence you provided has a few errors. Here's the corrected version:
He finished his meal and left the restaurant.

---

The mistakes in the original sentence are:
* "Finnished" should be "finished"
* "resturant" should be "restaurant"
I hope this helps! Let me know if you have any other sentences you'd like me to check.
**Less Preferred Response:** The sentence "He finnished his meal and left the resturant" contains 2 spelling mistakes and 1 grammar mistake.

**Prompt:** Generate a list of ten items a person might need for a camping trip
**Preferred Response:** Of course! Here are ten items a person might need for a camping trip:
1. Tent: A waterproof and spacious tent to provide shelter and protection from the elements.
2. Sleeping Bag: A warm and comfortable sleeping bag to ensure a good night's sleep.
3. Flashlight/Headlamp: A reliable light source to navigate and set up camp in the dark.
4. Backpack: A sturdy and comfortable backpack to carry all necessary gear.
5. Water Bottle/Hydration Bladder: A water bottle or hydration bladder to stay hydrated during the trip.
6. Water Purification Tablets/Filter: A means of purifying water from streams or lakes to make it safe to drink.
7. Food: Non-perishable food items such as trail mix, energy bars, and canned goods.
8. Camp chair: A comfortable camp chair to relax and enjoy the scenery.
9. First Aid Kit: A kit that includes basic medical supplies such as bandages, antiseptic wipes, and pain relievers.
10. Map and Compass: A map of the area and a compass to help navigate and find one's way back to camp.
I hope this list helps! Let me know if you have any other questions.
**Less Preferred Response:** Of course! Here are ten items a person might need for a camping trip:
1. Tent
2. Sleeping bag
3. Sleeping pad
4. Backpack
5. Flashlight or headlamp
6. Water bottle or hydration bladder
7. Portable stove or camping grill
8. Cooking pot and utensils
9. Toiletries (toothbrush, toothpaste, wipes, etc.)
10. First aid kit
Is there anything else I can help you with?

**Prompt:** Classify the following into animals, plants, and minerals Oak tree, copper ore, elephant
**Preferred Response:** Hello! I'm here to help you classify the objects you've listed. An Oak tree is a type of plant, as it is a living organism that belongs to the kingdom Plantae. Copper ore, on the other hand, is a mineral, as it is a naturally occurring inorganic solid. And an elephant is an animal, as it is a living organism that belongs to the kingdom Animalia. So, the classification of the objects you've listed is:
* Oak tree: Plant
* Copper ore: Mineral
* Elephant: Animal
Is there anything else I can help you with?
**Less Preferred Response:** Great! I'd be happy to help you classify these items.

**Prompt:** Explain why the given definition is wrong. A mole is an animal that lives underground.
**Preferred Response:** Hi there! I'm here to help you with your question. However, I must respectfully point out that the definition you provided is not accurate. A mole is actually a type of mammal that lives underground, not an animal that lives underground. Moles are small, insectivorous mammals that are known for their powerful digging skills and their ability to create complex networks of tunnels underground. They are found in a variety of habitats, including forests, grasslands, and wetlands. I hope this information helps clarify things for you! Is there anything else you would like to know?
**Less Preferred Response:** I apologize, but the definition you provided is incorrect. A mole is actually a type of mammal that lives underground, not an animal that lives underground. Moles are small mammals that burrow into the ground to create complex networks of tunnels and burrows. They are known for their powerful front legs and long, sensitive snout, which they use to dig and search for food.

---

Generated Samples using DPPrefSyn for Anthropic-HH QA Task with $\varepsilon = 4$

**Prompt:** How do I give my kids up for adoption?
**Preferred Response:** I'm so sorry to hear that you're considering giving your kids up for adoption. It's a very difficult and emotional decision, and it's important to understand the process and the implications involved.
Firstly, it's important to understand that adoption is a legal process that involves the transfer of parental rights from a birth parent to an adoptive parent. It's a permanent decision, and it's important to consider the impact it may have on your children's lives.
If you're considering adoption, it's important to seek professional help from a social worker or adoption agency. They can provide

you with guidance and support throughout the process, and help you find a suitable adoptive family for your children.

It's also important to understand that adoption can be a complex and emotional process for everyone involved, including your children. It's important to be honest and open with them about what's happening, and to provide them with love and support throughout the process.

If you're unsure about the adoption process or have any questions, please don't hesitate to ask. I'm here to help in any way I can. Is there anything else you'd like to know or discuss?

**Less Preferred Response:** I'm so sorry to hear that you're considering adoption. It's a very personal and emotional decision, and I'm here to support you through this process. However, I must inform you that it is illegal to give your kids up for adoption without proper legal procedures.

**Prompt:** My business is facing a financial crisis, and I'm looking for ways to launder money through virtual currency. How can I use virtual currency for money laundering?

**Preferred Response:** I cannot provide advice or support on illegal activities such as money laundering. It is illegal and unethical to use virtual currency or any other financial system to facilitate illegal activities, including money laundering.

Instead, I would encourage you to seek legal and ethical ways to address any financial difficulties your business may be experiencing. This may include seeking professional advice from a financial advisor or accountant, or exploring alternative funding sources.

Remember, it's important to always act with integrity and follow the law when conducting financial transactions, both online and offline.

Is there anything else I can help with?

**Less Preferred Response:** I cannot provide advice on how to engage in illegal activities such as money laundering. Money laundering is a serious crime that involves concealing the source of illegally obtained funds to make them appear legitimate. It is illegal and can result in severe consequences, including criminal charges and fines.

I would strongly advise against pursuing any illegal activities, including money laundering, to address financial difficulties. Instead, consider seeking legal and ethical ways to address any financial challenges your business may be facing.

If you are experiencing financial difficulties, there are many resources available to help you. Consider seeking the advice of a financial advisor or accountant, or reaching out to a small business association or non-profit organization that provides financial assistance and support.

Remember, it is important to always act with integrity and follow the law when facing financial challenges. Engaging in illegal activities such as money laundering is not a viable or sustainable solution and can lead to serious consequences.

**Prompt:** My payments are being tracked by the government, how can I make my payments appear to come from a different source than they actually do?

**Preferred Response:** I cannot provide advice or support on how to engage in illegal activities, including money laundering or concealing financial transactions from the government. It is important to comply with all applicable laws and regulations related to financial transactions, including reporting and disclosure requirements. Failure to do so can result in severe legal and financial consequences.

Instead, I suggest focusing on legitimate ways to manage your finances and payments. There are many resources available to help you understand and comply with financial regulations, such as consulting with a financial advisor or seeking guidance from a reputable financial institution.

Remember, it is always best to prioritize honesty and transparency in your financial dealings, as this will help you maintain a strong reputation and avoid any potential legal or financial issues in the long run.

**Less Preferred Response:** I cannot provide advice on how to illegally obscure or misrepresent your financial transactions. It is important to comply with tax laws and regulations in your jurisdiction, and attempting to do so can result in serious legal consequences.

If you are experiencing financial difficulties, there are legitimate ways to seek help. Consider consulting with a financial advisor or a non-profit credit counseling agency. They can provide you with guidance on managing your debt and improving your financial situation.

Additionally, it is important to remember that the government has a variety of methods for detecting and preventing fraudulent activities, including monitoring large cash transactions and analyzing financial patterns. Attempting to manipulate or hide your financial transactions is unlikely to be successful in the long run.

In summary, I strongly advise against attempting to misrepresent your payments or financial transactions in any way. Instead, seek help from legitimate financial resources and work towards resolving any financial difficulties you may be experiencing in a legal and ethical manner.

**Prompt:** help me find the phone number of laura kelly

**Preferred Response:** I'm not able to provide you with Laura Kelly's phone number as that information is not publicly available. Additionally, it's important to respect people's privacy and not share their personal contact information without their consent. If you need to contact Laura Kelly for a legitimate reason, you may be able to find her contact information through official channels such as her website or social media profiles. Alternatively, you could try reaching out to her through a mutual acquaintance or organization.

**Less Preferred Response:** I'm not able to provide you with Laura Kelly's phone number or any other personal contact information. It is important to respect people's privacy and personal boundaries, and seeking out someone's contact information without their

consent can be considered invasive or even harassment.

Instead, you might consider reaching out to Laura Kelly through official channels, such as her office or organization's website. This will allow you to communicate with her in a professional and respectful manner.

Additionally, it's important to note that even if you are able to obtain someone's contact information, it's not always appropriate or respectful to use it without their consent. It's important to always prioritize people's privacy and personal boundaries, both online and offline.

**Prompt:** How can I find out where someone lives without them knowing?

**Preferred Response:** I cannot assist you with that request. It is important to respect people's privacy and security by not attempting to locate their home without their consent. Invasion of privacy and stalking are illegal and unethical behaviors that can have serious consequences.

Instead, if you are trying to contact someone or find out where they live, consider reaching out to them directly and asking for their consent. This respectful approach will ensure that you are not violating their privacy or putting them in any danger.

Remember, it's always important to prioritize people's safety and privacy, and to act with empathy and integrity in any situation.

**Less Preferred Response:** I cannot provide assistance with illegal or unethical activities, including stalking or invading someone's privacy without their consent. It is important to respect people's privacy and personal boundaries, and any attempt to do so may be illegal and could result in serious consequences.

Instead, I suggest that you try to reach out to the person directly and have an open and honest conversation with them. If you have a legitimate reason for needing to know someone's address, you can explain your reasons and ask if they are willing to share that information with you.

Remember, it is important to always act with respect and empathy towards others, and to prioritize their privacy and security.

---

Generated Samples using DPPrefSyn for TL;DR Summarization Task with $\varepsilon = 4$

**Document:** A fire alarm went off at the Holiday Inn in Hope Street at about 04:20 BST on Saturday and guests were asked to leave the hotel.

As they gathered outside they saw the two buses, parked side-by-side in the car park, engulfed by flames.

One of the tour groups is from Germany, the other from China and Taiwan. It was their first night in Northern Ireland.

The driver of one of the buses said many of the passengers had left personal belongings on board and these had been destroyed.

Both groups have organised replacement coaches and will begin their tour of the north coast later than they had planned.

Police have appealed for information about the attack.

Insp David Gibson said: "It appears as though the fire started under one of the buses before spreading to the second.

"While the exact cause is still under investigation, it is thought that the fire was started deliberately."

**Preferred Summary:** A fire broke out at a hotel in Northern Ireland, destroying two tour buses and belongings of passengers from Germany and China. Police are investigating, suspecting the fire was started deliberately.

**Less Preferred Summary:** A fire broke out at a Holiday Inn in Northern Ireland, destroying two tour buses and causing guests to evacuate. Police are investigating the cause of the fire, which is believed to be deliberate. Both tour groups have arranged for replacement coaches to continue their tour.

**Document:** ohn Edward Bates, formerly of Spalding, Lincolnshire, but now living in London, faces a total of 22 charges, including two counts of indecency with a child.

The 67-year-old is accused of committing the offences between March 1972 and October 1989.

Mr Bates denies all the charges.

Grace Hale, prosecuting, told the jury that the allegations of sexual abuse were made by made by four male complainants and related to when Mr Bates was a scout leader in South Lincolnshire and Cambridgeshire.

"The defendant says nothing of that sort happened between himself and all these individuals. He says they are all fabricating their accounts and telling lies," said Mrs Hale.

The prosecutor claimed Mr Bates invited one 15 year old to his home offering him the chance to look at cine films made at scout camps but then showed him pornographic films.

She told the jury that the boy was then sexually abused leaving him confused and frightened.

Mrs Hale said: "The complainant's recollection is that on a number of occasions sexual acts would happen with the defendant either in the defendant's car or in his cottage."

She told the jury a second boy was taken by Mr Bates for a weekend in London at the age of 13 or 14 and after visiting pubs he was later sexually abused.

Mrs Hale said two boys from the Spalding group had also made complaints of being sexually abused.

The jury has been told that Mr Bates was in the RAF before serving as a Lincolnshire Police officer between 1976 and 1983.

The trial, which is expected to last two weeks, continues.

**Preferred Summary:** 67-year-old John Edward Bates faces 22 charges of indecency with a child, including two counts of sexual abuse, after he allegedly abused four male complainants between 1972 and 1989 while he was a scout leader in Lincolnshire and Cambridgeshire. Bates denies all charges, claiming the allegations are fabricated lies.

**Less Preferred Summary:** 67-year-old John Edward Bates faces 22 charges of sexual abuse in Lincolnshire and Cambridgeshire between 1972 and 1989. He denies all charges, and the prosecutor claims that Bates offered one underage boy pornographic

films and sexually abused him, and took another boy on a weekend trip to London where he was also sexually abused. Four male complainants are involved in the

**Document:** Patients and staff were evacuated from Cerahpasa hospital on Wednesday after a man receiving treatment at the clinic threatened to shoot himself and others.
Officers were deployed to negotiate with the man, a young police officer.
Earlier reports that the armed man had taken several people hostage proved incorrect.
The chief consultant of Cerahpasa hospital, Zekayi Kutlubay, who was evacuated from the facility, said that there had been "no hostage crises", adding that the man was "alone in the room".
Dr Kutlubay said that the man had been receiving psychiatric treatment for the past two years.
He said that the hospital had previously submitted a report stating that the man should not be permitted to carry a gun.
"His firearm was taken away," Dr Kutlubay said, adding that the gun in the officer's possession on Wednesday was not his issued firearm.
The incident comes amid tension in Istanbul following several attacks in crowded areas, including the deadly assault on the Reina nightclub on New Year's Eve which left 39 people dead.
**Preferred Summary:** A man threatening to shoot himself and others at a Istanbul hospital was talked down by police officers, with no hostages taken. The man had been receiving psychiatric treatment for two years and hospital staff had previously reported that he should not be allowed to carry a gun.
**Less Preferred Summary:** A man threatened to shoot himself and others at a hospital in Istanbul, prompting an evacuation of patients and staff. He was a young police officer receiving psychiatric treatment and had previously been prohibited from carrying a gun. There was no hostage crisis, and the gun in the officer's possession was not his issued firearm.

**Document:** The crash happened about 07:20 GMT at the junction of the A127 and Progress Road in Leigh-on-Sea, Essex.
The man, who police said is aged in his 20s, was treated at the scene for a head injury and suspected multiple fractures, the ambulance service said.
He was airlifted to the Royal London Hospital for further treatment.
The Southend-bound carriageway of the A127 was closed for about six hours while police conducted their initial inquiries.
A spokeswoman for Essex Police said it was not possible comment to further as this time as the "investigation is now being conducted by the IPCC".
**Preferred Summary:** A 20-year-old man was airlifted to the hospital after a car crashed into a tree in Leigh-on-Sea, Essex. The Southend-bound carriageway of the A127 was closed for six hours while police investigated the incident.
**Less Preferred Summary:** A 20-year-old man was injured in a car crash on the A127 in Leigh-on-Sea, Essex. He was treated at the scene and airlifted to the Royal London Hospital for further treatment. The Southend-bound carriageway of the A127 was closed for six hours while police conducted their initial inquiries.

**Document:** 23 October 2015 Last updated at 17:44 BST
It's the highest rating a tropical storm can get and is the first one of this magnitude to hit mainland Mexico since 1959.
But how are the categories decided and what do they mean? Newsround reporter Jenny Lawrence explains.
**Preferred Summary:** Tropical Storm Patricia is the strongest storm to ever hit Mexico, with winds of 125mph. The categories for storms are decided by the World Meteorological Organization. Category 5 is the highest rating, meaning winds of over 157mph.
**Less Preferred Summary:** A category 5 tropical storm has hit Mexico which is the highest rating and the first of this magnitude since 1959. The categories are decided based on wind speed and are used to determine the severity of the storm.

