# OpenReview forum: "Differentially Private Preference Data Synthesis for Large Language Model Alignment"
_ICML.cc/2026/Conference — ICML 2026 regular_

### Official Review · Reviewer_Bsfi · 2026-02-21

**Soundness:** 2
**Presentation:** 3
**Significance:** 2
**Originality:** 2
**Overall Recommendation:** 3
**Confidence:** 3

**Summary:**

To address the privacy concerns in LLMs preference alignment, this paper proposes a differential privacy method based on data synthesis, called DPPrefSyn. Starting with a private dataset, the method first trains reward models for different clusters using embedding difference, DP-PCA dimensionality reduction, DP-KMeans, and DP-SGD. Then, the method samples from a public dataset and uses the reward models to score responses, thereby constructing a preference dataset.

**Compliance With Llm Reviewing Policy:**

Affirmed.

**Final Justification:**

Refer to the reply

**Key Questions For Authors:**

Refer to Weakness.

**Limitations:**

yes

**Strengths And Weaknesses:**

Strengths:
1. The paper presents a clear logical structure, and the proposed idea of ​​DPPrefSyn is easy to follow.

2. DPPrefSyn follows the preference modeling of Bradley–Terry (BT) and provides a feasible solution for obtaining privacy preferences across multiple dimensions through DP-KMeans.

3. The authors conducted a theoretical analysis to ensure that the proposed method follows the constraints of DP.

Weakness:
1. The motivation behind this paper is ambiguous. While it lists some limitations of existing differential privacy methods, the authors do not explicitly explain how DPPrefSyn addresses these issues. I think the proposed method aims only to resolve the main contradictions of DP, not to improve existing solutions.

2. As shown in Fig. 1, this method performs K-Means on the embedding difference. However, I believe the purpose of DP-KMeans is to obtain privacy preferences across different dimensions, whereas the Embedding Difference itself removes these differences. So, why not perform K-Means first and then take the difference?

3. The main issue with the paper is that it fails to prove, either experimentally or theoretically, that DPPrefSyn can effectively address the biases between private and public prompts, even though the reward model has limitations in generalization.

4. The authors selected only one baseline for comparison and did not include common differential privacy methods.

---

> ### Author Rebuttal · Authors · 2026-03-31
>
> We thank the reviewer for the careful reading and thoughtful comments. We hope the responses below address the reviewer's concerns.
>
> **Q1:** How does DPPrefSyn address limitations of existing methods?
>
> **A1:** Existing methods have three key limitations that DPPrefSyn addresses:
>
> 1. **Partial privacy protection.** Chowdhury et al. (2024) and Zhang et al. (2025) protect only preference labels using a weaker DP notion (label-DP), leaving prompts and responses unprotected. DPPrefSyn protects the **entire preference sample** under a stronger DP guarantee.
>
> 2. **Method-specific.** Wu et al. (2023a) are limited to PPO. DPPrefSyn generates **reusable synthetic data** compatible with any preference alignment method (SFT, DPO, PPO).
>
> 3. **Constrained by limited private data.** Existing methods rely on scarce human preference annotations. DPPrefSyn overcomes this by decoupling preference modeling from data generation, scaling far beyond the private dataset size.
>
> We will make these connections more explicit in the revision.
>
> **Q2:** Why compute embedding difference first, then cluster? Why not cluster first, then take the difference?
>
> **A2:** The embedding difference does not remove preference information, in fact, it is **the quantity that reveals preference patterns** under the Bradley-Terry model. Specifically, $P[a_i^+ \succ a_i^- | x_i] = \sigma(\langle \theta, \phi(x_i, a_i^+) - \phi(x_i, a_i^-) \rangle)$. For a preference to be strong (high probability), the preference vector $\theta$ must be strongly aligned with the embedding difference $\phi(x_i, a_i^+) - \phi(x_i, a_i^-)$. Therefore, for annotators with similar preferences, the embedding differences across their rated pairs are expected to point in similar directions in the embedding space. We cluster on these difference vectors so preferences within each cluster are well approximated by a shared $\theta$.
>
> If we clustered the raw embeddings first (before taking differences), we would group samples by **topic similarity** (e.g., all health-related samples together), not by **preference similarity**. Since our goal is to capture diverse preference patterns, the difference must come first.
>
> **Q3:** It fails to prove that DPPrefSyn can effectively address the biases between private and public prompts.
>
> **A3:** We provide both a conceptual argument and empirical evidence.
>
> **Conceptual argument.** In DPPrefSyn, preferences are captured by $\theta_k$, the parameter in the reward model, not $\phi(x,a)$, the embedding of the prompt-response pair. An annotator may interact with an LLM through diverse prompts, but their underlying preference does not change across different prompts. For example, an annotator who prefers concise responses will have the same $\theta_k$ whether the prompt is about health or coding. Therefore, a reward model trained on private prompts can generalize to public prompts, as the preference pattern it captures is **prompt-independent**.
>
> **Empirical evidence.**
>
> 1. Cross-domain evaluation (Table 1). Our experimental results show effectiveness under real prompt distribution shifts. Specifically, we use Alpaca prompts for OpenAssistant, SafeRLHF prompts for Anthropic-HH, and XSum prompts for TL;DR, all of which are distributionally different from the private datasets. The consistently strong performance provides empirical validation.
>
> 2. Controlled experiment (Table 13). We compare DPPrefSyn using Alpaca prompts (with distribution shift) against using the same OpenAssistant prompts as the private data (no shift). We find that performance remains strong in both cases, and distribution shift does not hurt performance.
>
> **Q4:** Only one baseline is included; common DP methods are missing.
>
> **A4:** We clarify why direct comparison with existing methods is not straightforward.
>
> - Chowdhury et al. (2024) and Zhang et al. (2025) adopt a weaker DP notion (label-DP) that protects only preference labels, not prompts or responses. As they use a different DP notion, a direct comparison would not be meaningful.
>
> - Wu et al. (2023a) are limited to PPO and cannot be applied to DPO.
>
> - General DP synthetic text methods like Aug-PE (Xie et al., 2024) are designed for unstructured text. Preference data synthesis requires not only generating text but also modeling human preferences to determine which response is preferred. These methods have no mechanism for learning such preferences, so they cannot produce valid preference pairs. Adapting these methods to the preference setting is non-trivial and would require significant modifications, making direct comparison unfair.
>
> DP-FT is a strong natural baseline: it uses DP-SGD, the most widely-used DP training method. DPPrefSyn not only outperforms DP-FT, but also surpasses non-private fine-tuning on real data.
>
> ---
>
> We thank the reviewer again for the helpful comments. If our responses resolve your concerns, we kindly request the reviewer to consider raising the rating. We are happy to address any further questions.

---

> > ### Author Rebuttal · Reviewer_Bsfi · 2026-04-03
> >
> > Thank you for the response. I have adjusted my score based on the rebuttal. However, I still believe the method needs more baseline comparisons to demonstrate its reliability.

---

> > > ### Author Response · Authors · 2026-04-04
> > >
> > > We thank the reviewer for the engagement and for raising the score. Following the reviewer's suggestion, we add a comparison with DP-PPO (Wu et al., 2023a), which applies DP-SGD to different stages of RLHF. However, it is limited to PPO and cannot be applied to other alignment methods such as DPO, while DPPrefSyn generates reusable synthetic data that can be applied to multiple preference alignment methods.
> > >
> > > Since Wu et al. (2023a) do not release their code, we adopt their reported results from Table 3 and align with their task setup and evaluation protocol to ensure a fair comparison. Specifically, we evaluate on the TL;DR summarization task using GPT-2 Large, and compute ROUGE scores (ROUGE-1, ROUGE-2, and ROUGE-L) between model-generated summaries and reference summaries in the test set, as done in Wu et al. (2023a). The results are shown below.
> > >
> > > |   | ROUGE-1 | ROUGE-2 | ROUGE-L |
> > > |---|---|---|---|
> > > | $\varepsilon=0$ | 16.13 | 1.56 | 10.34 |
> > > | DP-PPO ($\varepsilon=4$) | 19.63 | 2.83 | 13.88 |
> > > | **DPPrefSyn ($\varepsilon=4$)** | **24.02** | **6.80** | **18.28** |
> > >
> > > We find that **DPPrefSyn significantly outperforms DP-PPO alignment**. Moreover, the synthetic data generated by DPPrefSyn can be **reused across various alignment methods and models** without incurring additional privacy cost.
> > >
> > > For other methods, as discussed in our previous response, direct comparison is not straightforward: label-DP methods (Chowdhury et al., 2024; Zhang et al., 2025) use a different privacy definition, and general DP synthetic text methods (Xie et al., 2024) have no mechanism for modeling human preferences and cannot produce valid preference pairs.
> > >
> > > We hope this addresses the reviewer's concern. We are happy to include any further comparisons the reviewer considers appropriate. If our response resolves your concerns to a satisfactory level, we kindly request the reviewer to consider raising the rating of our work.

---

### Official Review · Reviewer_akAh · 2026-03-09

**Soundness:** 3
**Presentation:** 3
**Significance:** 2
**Originality:** 3
**Overall Recommendation:** 4
**Confidence:** 4

**Summary:**

Preference alignment is attracting growing attention, and the authors propose a novel solution using DP techniques to generate synthetic data under privacy constraints. The solution consists of three stages: Step 1 uses an embedding model to compute the differences for each preference pair, which serve as features; then it proposes a DP-PCA method to reduce the dimension to p, and applies K-means to cluster the samples. Step 2 trains a linear reward model \theta on each cluster with DP-SGD. Step 3 uses the DP reward model to rank responses generated on public prompts to form a DP synthetic preference dataset. The resulting dataset can then be used for downstream tasks with other optimization algorithms.

**Compliance With Llm Reviewing Policy:**

Affirmed.

**Final Justification:**

In the rebuttal period, after carefully reading the discussions between the authors and the reviewers, I think most of my concerns regarding the paper have been addressed. While I may still respectfully retain my opinion regarding the contribution of the work (as I mentioned in the initial review), I agree that the method works well, and the quality of the work could be greatly improved with comments from all reviewers. Therefore, I would like to adjust my rating.

**Key Questions For Authors:**

Please see the weakness part above.

**Limitations:**

yes

**Strengths And Weaknesses:**

## Strengths

* The solution proposed by the authors involves multiple DP approaches, such as DP-PCA, DP k-means, and DP-SGD, which are classical methods and well understood by readers familiar with privacy. The pipeline is reasonable and gives readers intuitive confidence in the validity of the approach.

* Instead of directly converting an optimization algorithm such as DPO into a DP version, the authors take an interesting detour: they build DP synthetic data indirectly from a private preference dataset, and manage the privacy budget consumed in the generation pipeline well.

* The overall writing is well-organized and easy to follow.

## Weakness

* The combination of DP approaches forms a valid solution for generating a preference synthetic dataset under DP. However, these approaches are so classical that it has two effects. On the one hand, the overall combination feels quite straightforward and very easy to follow. On the other hand, it also makes the work feel more like nice engineering rather than solid research work.

* While the authors explicitly mention weaknesses of other work in the related work section, the best justification would be to fix some total \epsilon and show the performance differences—this seems to be missing in the experiment part. The authors could at least explain why such a comparison is not included (e.g., other dp synthetic data generation method) - DP-FT might be a too-weak baseline (given the well-known limitations of DP-SGD).

* Also, I am very curious how the split of the privacy budget is set in your experiments; i.e., how are \epsilon_0 and \epsilon_1 chosen given a total \epsilon? Intuitively, this may play an important role in final performance. If it does not, please provide some explanation.

* While several small models are tested in the experiment, I am curious how the scaling laws behave in this case, both in terms of model size and data size (also, difficulty). But I would not insist on this, as resources might be a crucial limitation for the authors.

---

> ### Author Rebuttal · Authors · 2026-03-31
>
> We thank the reviewer for the careful reading and thoughtful comments. We hope the responses below address the reviewer's concerns.
>
> **Q1:** The combination of DP approaches feels quite straightforward and makes the work feel more like nice engineering rather than solid research work.
>
> **A1:** While the individual DP components (DP-PCA, DP-KMeans, DP-SGD) are classical, the key contribution is the insight that connects them into a principled framework for preference data synthesis. This insight is non-trivial and grounded in the structure of the problem:
>
> 1. **The connection between the Bradley-Terry model, linear rewards, and clustering.** Under the Bradley-Terry model with a linear reward, the preference probability depends on $\langle \theta, \phi(x, a^+) - \phi(x, a^-) \rangle$. For annotators with similar preferences, the embedding differences $\phi(x, a^+) - \phi(x, a^-)$ across their rated pairs are expected to point in similar directions. We can therefore cluster on these difference vectors so that preferences within each cluster are well approximated by a shared $\theta_k$. Without this insight, it would be unclear *what* to cluster or *why* clustering helps.
>
> 2. **The use of public prompts to decouple preference modeling from prompt synthesis.** Generating prompts, preferred responses, and dispreferred responses simultaneously under DP would quickly exhaust the privacy budget. Our key observation is that user preferences are captured by $\theta_k$, not by the prompts themselves, because a user's underlying preference does not change across different prompts. This allows us to spend the entire privacy budget on modeling preferences and use public prompts freely for generation.
>
> 3. **The resulting framework has unique properties that no prior work achieves.** DPPrefSyn protects both prompts and preference labels simultaneously, is compatible with multiple alignment methods (SFT, DPO, PPO), and produces reusable synthetic data, all under formal DP guarantees. No existing method offers this combination (see A2).
>
> Our technical novelty lies in identifying the right problem structure and connecting it to the right tools, which is a research contribution beyond engineering.
>
> **Q2:** Comparison with other DP methods (e.g., other DP synthetic data generation methods) is missing in the experiment part.
>
> **A2:** We would like to clarify why direct comparison with existing methods is not straightforward:
>
> - Chowdhury et al. (2024) and Zhang et al. (2025) adopt a weaker DP notion (label-DP) that protects only preference labels, not prompts or responses.
>
> - Wu et al. (2023a) are limited to PPO and cannot be applied to DPO.
>
> - General DP synthetic text methods like Aug-PE (Xie et al., 2024) are designed for unstructured text. Preference data synthesis requires not only generating text but also modeling human preferences to determine which response is preferred. These methods have no mechanism for learning such preferences, so they cannot produce valid preference pairs. This is exactly the challenge DPPrefSyn addresses through its reward models. Adapting prior methods to the preference setting is non-trivial and would require significant modifications, making it an unfair comparison.
>
> **Q3:** How are $\varepsilon_0$ and $\varepsilon_1$ chosen? Does the split matter?
>
> **A3:** The core principle of our budget allocation is to reserve the majority of the budget for DP-SGD reward model training, since this step directly determines the quality of preference labels in the synthetic data. In our experiments, we allocate $\varepsilon_0=\varepsilon_1=\varepsilon/8$ by default, reserving 75\% of the budget for reward model training.
>
> We verify this choice in Table 12 (Appendix B.5) by evaluating three different allocations under $\varepsilon=4$: (1) the default setting ($\varepsilon_0=\varepsilon_1=0.5$), (2) allocating more budget to DP-PCA ($\varepsilon_0=0.7,\varepsilon_1=0.3$), and (3) allocating more budget to DP-KMeans ($\varepsilon_0=0.3,\varepsilon_1=0.7$). We find that performance varies by less than 0.4\% after DPO, showing that DPPrefSyn is robust to reasonable changes in the budget split.
>
> **Q4:** Scaling behavior with model size and data size.
>
> **A4:** Our current experiments cover models from 2B to 8B parameters and data sizes from 52K to 204K, and DPPrefSyn consistently performs well across these settings. We agree that a comprehensive scaling law study would be valuable. Due to computational resource constraints, we were unable to conduct experiments at larger scales (e.g., 70B) in this submission. We believe investigating the scaling behavior of DPPrefSyn is a promising and important direction for future work.
>
> ---
>
> We thank the reviewer again for the helpful comments. If our responses resolve your concerns, we kindly request the reviewer to consider raising the rating. We are happy to address any further questions.

---

> > ### Author Rebuttal · Reviewer_akAh · 2026-04-02
> >
> > I am always happy to see solid works in the LLM regime that puts privacy into consideration. However, I may take the opinion regarding the definition of DP for preferences with a grain of salt, and I may respectfully keep my original rating. Thanks again.

---

> > > ### Author Response · Authors · 2026-04-04
> > >
> > > We thank the reviewer for the continued engagement and appreciate the feedback on this point.
> > >
> > > We would like to clarify that our use of differential privacy (DP) for privatizing preferences is consistent with the established DP preference alignment literature [1–3]. These works share the common goal of applying DP to protect sensitive information in preference alignment, differing in what is protected and how. [1] adopts the same DP definition used in our work, protecting prompts, responses, and preference labels. [2] and [3] adopt the weaker label-DP notion, protecting only preference labels. Our work follows [1], providing comprehensive protection covering the entire preference sample.
> > >
> > > The distinction from [1] lies in the approach rather than the DP formulation. [1] applies DP-SGD to directly fine-tune a language model using PPO, providing privacy guarantees over the RLHF training process. In contrast, our method privatizes the preference data itself by generating reusable synthetic preferences, which can then be applied to multiple downstream alignment objectives without re-privatizing for each one. This offers a more modular and data-efficient alternative while maintaining the same formal DP guarantees.
> > >
> > > We therefore believe our DP definition is well-grounded in the literature. That said, we would appreciate it if the reviewer could point to the specific aspect of the DP definition that raises concern, whether it relates to the privacy accounting, the threat model, or another dimension, so that we can address it precisely in the revision.
> > >
> > > [1] Wu, Fan, et al. "Privately Aligning Language Models with Reinforcement Learning." The Twelfth International Conference on Learning Representations.
> > >
> > > [2] Chowdhury, Sayak Ray, Xingyu Zhou, and Nagarajan Natarajan. "Differentially private reward estimation with preference feedback." International Conference on Artificial Intelligence and Statistics. PMLR, 2024.
> > >
> > > [3] Zhang, Jiaming, et al. "Towards User-level Private Reinforcement Learning with Human Feedback." Second Conference on Language Modeling.

---

### Official Review · Reviewer_yKag · 2026-03-12

**Soundness:** 3
**Presentation:** 3
**Significance:** 3
**Originality:** 3
**Overall Recommendation:** 5
**Confidence:** 2

**Summary:**

The authors propose a three-step methodology for differentially private training. First, the prompts and answers from the private dataset are embedded, followed by the application of DP-PCA and DP-K-Means. Second, DP-SGD is used to train a linear reward model based on these differentially private representations. Finally, an LLM uses public prompts to sample multiple generations; these are embedded and scored by the linear reward model to construct a synthetic set of preferences. The authors show the effectiveness of their method across three datasets (OpenAssistant, Anthropic-HH, and TL;DR).

**Compliance With Llm Reviewing Policy:**

Affirmed.

**Final Justification:**

I feel that the authors have addressed and clarified all my concerns, and hence decided to increase my score.

**Key Questions For Authors:**

**Q1:** While I understand that the authors perform K-Means clustering, in order to account for different behavioural modes in the preference, and they show this in Figure 2a, I wonder how much it actually is able to separate modes? E.g. do the clusters represent helpfulness and harmlessness in the case of anthropic HH? Also, why $K=5$?

**Limitations:**

The authors do not include a dedicated limitations section, and I would encourage them to provide more detail. Possible points of discussion could include the increased computational resources required by their method compared to other DP methods, or the loss in performance.

**Strengths And Weaknesses:**

Overall, I think this is an interesting approach and well done in terms of the soundness of their experiments. However, I wish to point out that I am not very familiar with differential privacy or the related work, but only familiar with preference learning itself. Therefore, I give only a low confidence score and hope that the Area Chair takes this into consideration.

# Strengths:
**S1:** I believe that the problem of preference learning is being investigated a lot nowadays, but I appreciate the authors' angle on differentiable private methods, as this could indeed be a problem with collected user data

**S2:** I appreciate that the authors report the standard deviation over five runs. These error bars give a clear picture of how stable their method is

**S3:** Generally, I feel the paper is well written, and the research questions are addressed with ample experiments

# Weaknesses:
**W1:**: Not all results have error bars, such as Table 5 and Figure 2. I would encourage the authors to provide this

**W2:** I would encourage the authors to use a different baseline for the win-rates, rather than testing against gpt-4o, as this often yields very low scores. Therefore, I would recommend testing, e.g. the win-rate of DP-FT vs DPPrefSyn directly, as that makes it more comparable, rather than going through a proxy baseline for GPT-4o

---

> ### Author Rebuttal · Authors · 2026-03-31
>
> We thank the reviewer for the careful reading and thoughtful comments. We hope the responses below address the reviewer's concerns.
>
> **W1:** Not all results have error bars, such as Table 5 and Figure 2.
>
> **Response:** We have computed the standard deviations over 5 random seeds and report them below.
>
> For Table 5:
>
> | Downstream LLM | Method | Generator LLM | $\varepsilon=4$ | $\varepsilon=\infty$ |
> |---|---|---|---|---|
> | | DP-FT | - | $70.15\pm 2.09$ | $73.52\pm 2.30$ |
> | Qwen-3-4B | DPPrefSyn | Qwen-3-4B-Instruct | $50.99\pm 1.10$ | $50.87\pm 0.84$ |
> | | DPPrefSyn | Llama-3-8B-Instruct | $50.64\pm 1.29$ | $49.98\pm 1.19$ |
> | | DP-FT | - | $62.88\pm 1.78$ | $65.39\pm 2.59$ |
> | Gemma-2-2B | DPPrefSyn | Qwen-3-4B-Instruct | $50.51\pm 1.38$ | $50.43\pm 0.91$ |
> | | DPPrefSyn | Llama-3-8B-Instruct | $50.77\pm 0.83$ | $51.10\pm 0.54$ |
> | | DP-FT | - | $61.95\pm 2.05$ | $70.13\pm 1.24$ |
> | Llama-3-8B | DPPrefSyn | Qwen-3-4B-Instruct | $50.53\pm 0.62$ | $50.96\pm 0.95$ |
> | | DPPrefSyn | Llama-3-8B-Instruct | $50.45\pm 0.87$ | $50.67\pm 0.96$ |
>
> For Figure 2a:
>
> | Variant | SFT | SFT+DPO |
> |---|---|---|
> | w/o DP-KMeans | $7.78\pm 0.68$ | $10.31\pm 0.62$ |
> | w/o DP-PCA | $8.65\pm 0.42$ | $10.50\pm 0.61$ |
> | DPPrefSyn | $9.33\pm 0.56$ | $11.85\pm 0.99$ |
>
> For Figure 2b:
> | K | SFT | SFT+DPO |
> |---|---|---|
> | 1 | $7.78\pm 0.68$ | $10.31\pm 0.62$ |
> | 3 | $9.05\pm 0.48$ | $10.93\pm 0.50$ |
> | 5 | $9.33\pm 0.56$ | $11.85\pm 0.99$ |
> | 8 | $9.89\pm 0.34$ | $11.29\pm 0.49$ |
>
> We will include these in the revision.
>
> **W2:** I would recommend testing, e.g. the win-rate of DP-FT vs DPPrefSyn directly.
>
> **Response:** Following the reviewer's recommendation, we conduct a direct comparison where GPT-4o judges DPPrefSyn's outputs against DP-FT's outputs on OpenAssistant. We report the win rate of DPPrefSyn over DP-FT below.
>
> | $\varepsilon$ | 0.5 | 1 | 2 | 4 | 8 | $\infty$ |
> |---|---|---|---|---|---|---|
> | DPPrefSyn win rate (\%) | 69.94 | 71.91 | 72.75 | 73.46 | 72.89 | 74.29 |
>
> DPPrefSyn is preferred over DP-FT in over 70\% of cases across different privacy budgets, confirming the consistent advantage observed in our main results. We are running experiments on other datasets with multiple random seeds and will include the full results in the revision.
>
> **Q1:** I wonder how much clustering is able to separate modes? E.g. do the clusters represent helpfulness and harmlessness in the case of anthropic HH? Also, why $K=5$?
>
> **Response:** We provide a cluster interpretability analysis in Appendix B.8. Since the datasets used in our experiments do not include annotator group information, we design a controlled experiment following Chakraborty et al. (2024). We use GPT-4 to simulate two annotator groups with distinct preferences: one preferring responses understandable by elementary school students, the other preferring PhD-level responses. We then apply KMeans with $K=2$ to cluster the embedding differences. The resulting clusters strongly align with the simulated preference groups: one cluster is 95.6\% PhD-student preferences, the other is 93.1\% elementary-student preferences.
>
> Since annotator metadata is unavailable for Anthropic-HH, we cannot directly verify cluster correspondence. However, the controlled experiment demonstrates that our clustering can effectively separate distinct preference modes when they exist in the data.
>
> In DPPrefSyn, we treat $K$ as a hyperparameter. In Figure 2b, we conduct an ablation study with $K=1,3,5,8$. Performance improves from $K=1$ to $K=3$, and remains relatively stable at $K=5$ and $K=8$. This suggests that (1) capturing preference diversity with multiple clusters is beneficial over a single cluster, and (2) DPPrefSyn is robust to the exact choice of $K$, as long as it is not too small. We choose $K=5$ as a default that balances preference diversity with per-cluster sample size, as too many clusters would reduce the number of samples per cluster, making DP-SGD training less effective.
>
> **Q2:** No limitations section.
>
> **Response:** We will add a dedicated limitations section in the revision. DPPrefSyn requires LLM API access to synthesize preference data, which may incur additional API costs. Our work focuses on English text-based preference data, and extending DPPrefSyn to multilingual settings and multimodal data (e.g., image or video preferences) remains an open direction. Additionally, our experiments cover models up to 8B parameters due to computational constraints.
>
> ---
>
> We thank the reviewer again for the helpful comments. If our responses resolve your concerns, we kindly request the reviewer to consider raising the rating. We are happy to address any further questions.

---

> > ### Author Rebuttal · Reviewer_yKag · 2026-04-01
> >
> > I thank the authors for their detailed responses to my questions.
> >
> > I see my concerns as addressed, and therefore, I increase my score.
> >
> > However, I wish to reiterate that I am not too familiar with differential privacy, but rather with preference learning itself. I hope that the Area Chair takes this into consideration.

---

> > > ### Author Response · Authors · 2026-04-01
> > >
> > > Thank you very much for your positive feedback! Your valuable input has helped us improve the quality of our work significantly. We really appreciate it.

---

### Official Review · Reviewer_kEMy · 2026-03-13

**Soundness:** 2
**Presentation:** 3
**Significance:** 3
**Originality:** 2
**Overall Recommendation:** 5
**Confidence:** 3

**Summary:**

This paper tackles privacy concerns in preference alignment for large language models (LLMs), where human prompts and feedback may reveal sensitive information. Previous approaches using differential privacy (DP) often protect only part of the data and tend to be limited to specific methods like RLHF. The authors introduce DPPrefSyn, a framework for creating DP synthetic preference data based on established models (Bradley–Terry, DP-PCA, DP-KMeans, DP-SGD), allowing effective alignment for both RLHF and DPO while ensuring strong privacy guarantees. Experimental results confirm that DPPrefSyn’s synthetic data maintains privacy and achieves competitive alignment utility across tasks, offering a practical solution adaptable to emerging preference alignment techniques.

**Compliance With Llm Reviewing Policy:**

Affirmed.

**Final Justification:**

The author's response has resolved all of my concerns, and I am keeping my score as accept.

**Key Questions For Authors:**

Please refer to weaknesses.

**Limitations:**

There is no section discussing limitations; the paper would benefit from explicitly addressing potential weaknesses and open challenges.

**Strengths And Weaknesses:**

Strengths:
The paper presents a method built upon several well-established modules with formal guarantees, such as the DP mechanism and the Bradley–Terry preference model. By leveraging these mature components, the approach demonstrates a strong connection to prior advances in the field. Applying these techniques to the preference alignment problem is both rigorous and innovative, offering a promising direction for future research and applications.
The paper is clearly written, with well-structured explanations and thorough motivation for each step of the method.
Experiments are comprehensive, evaluating the approach across diverse scenarios and providing detailed analysis of the results.

Weaknesses:
The method assumes that human preferences can be effectively captured and clustered using a linear reward structure, which may oversimplify the diversity and complexity of real-world preference patterns. If annotator preferences are more nuanced or non-linear, clustering based solely on feature difference vectors might fail to accurately represent their true preferences, potentially limiting the effectiveness and generalizability of the synthetic data.
While DPPrefSyn outperforms fine-tuning on real data, this may reflect synthetic data being easier for models to learn and not fully capturing the complexity of real human preferences. The observed gains might not generalize to more diverse, real-world scenarios.
Although DPPrefSyn shows improved robustness against membership inference attacks, the evaluation is limited to a specific attack type and dataset. Its effectiveness against other privacy risks and stronger attack strategies remains unclear, raising questions about the generalizability of its empirical privacy protection. For instance, perform privacy risk evaluations (e.g. MIA) on multiple datasets with different characteristics, especially those closer to real-world, noisy, or ambiguous human preference data, to check the robustness and generality of the method.

---

> ### Author Rebuttal · Authors · 2026-03-31
>
> We thank the reviewer for the careful reading and thoughtful comments. We hope the responses below address the reviewer's concerns.
>
> **Q1:** Linear reward structure may oversimplify diverse and complex real-world preference patterns.
>
> **A1:** The linear reward assumption follows the standard framework in the preference alignment literature (Saha et al., 2023; Kong \& Yang, 2022; Zhu et al., 2023; Xiong et al., 2023). Given a preference dataset $\mathcal{D} = \\{(x_i, a_i^+, a_i^-)\\}\_{i=1}^n$, the reward model is commonly assumed to be linear in a feature map $r_\theta (x,a)=\langle\theta, \phi(x,a)\rangle$, where $\phi:\mathcal{X}\times\mathcal{A}\rightarrow\mathbb{R}^d$ is a known mapping and $\theta \in \mathbb{R}^d$ is a learnable parameter.
>
> Importantly, the "linearity" refers **only to the last layer**. In DPPrefSyn, $\phi(x,a)$ is the output of a pretrained deep sentence encoder (e.g., bge-large-en-v1.5) that captures highly nonlinear and complex semantic features. The overall mapping from raw text to reward values is therefore highly nonlinear. This is analogous to
> linear probing, where a linear head on top of strong pretrained features is known to be surprisingly expressive [1].
>
> Our framework further enhances expressiveness through **multiple preference clusters**, each with its own linear head $\theta_k$, which can better approximate diverse preference patterns.
>
> In Appendix B.6, we empirically compare our linear-head model with a fully fine-tuned GPT-2 reward model on Anthropic-HH. Our results show that **with strong pretrained features, a linear head is not only sufficient but competitive with end-to-end nonlinear alternatives**.
>
> Furthermore, Appendix B.8 shows that **our clustering recovers meaningful preference groups with high purity**, supporting that clustering feature difference vectors can capture diverse preference styles.
>
> [1] Alain, Guillaume, and Yoshua Bengio. "Understanding intermediate layers using linear classifier probes." arXiv preprint arXiv:1610.01644 (2016).
>
> **Q2:** DPPrefSyn outperforming fine-tuning on real data may reflect synthetic data being easier to learn rather than capturing real preferences.
>
> **A2:** We conduct controlled ablations in Appendix B.7 to test whether DPPrefSyn's gains come from prompt or response quality differences.
>
> In Table 13, we **fix the prompts** by giving DPPrefSyn the same OpenAssistant prompts used by DP-FT. We find that DPPrefSyn still outperforms DP-FT (10.39\% vs 6.12\%), indicating that its gains do not solely come from prompt quality differences.
>
> In Table 14, we **control response quality** by constructing an enhanced DP-FT baseline, which receives responses of comparable quality to those used in DPPrefSyn. Specifically, we use the same generator (LLaMA-7B-Chat) to produce candidate responses, rank them using a trained reward model, and apply DP-FT to this strengthened synthetic dataset. Under this controlled setting, DPPrefSyn outperforms the enhanced DP-FT baseline (11.85\% vs. 9.52\%), showing that its gains do not simply arise from response quality differences.
>
> These ablations isolate data quality as a confound, confirming that DPPrefSyn's gains come from its preference modeling component rather than data quality differences.
>
> **Q3:** MIA evaluation is limited to one attack type and dataset.
>
> **A3:** We note that the MIA we evaluate follows the state-of-the-art attack instantiated specifically for preference alignment data (Feng et al., 2025), making it a meaningful and relevant benchmark.
>
> We also highlight that DPPrefSyn's privacy guarantee is formal and theoretical, and this guarantee holds universally regardless of the attack strategy. MIA evaluation serves as complementary empirical validation rather than the basis of our privacy claim.
>
> We also run additional MIA experiments on Anthropic-HH and TL;DR (Qwen-3-4B as downstream LLM, Qwen-3-4B-Instruct as generator, $\varepsilon=4$):
>
> | Dataset | OpenAssistant | Anthropic-HH | TL;DR |
> |---|---|---|---|
> | DP-FT | 69.50\% | 56.86\% | 53.63\% |
> | DPPrefSyn | 50.92\% | 49.74\% | 49.62\% |
>
> We find that DPPrefSyn consistently achieves AUC scores close to random guessing ($\approx50\\%$), while DP-FT leads to  higher AUC scores. DPPrefSyn is thus more robust to empirical privacy attacks across different datasets and tasks.
>
> **Q4:** No limitations section.
>
> **A4:** We will add a dedicated limitations section in the revision. DPPrefSyn requires LLM API access to synthesize preference data, which may incur additional API costs. Our work focuses on English text-based preference data, and extending DPPrefSyn to multilingual settings and multimodal data remains an open direction. Additionally, our experiments cover models up to 8B parameters due to computational constraints.
>
> ---
>
> We thank the reviewer again for the helpful comments. If our responses resolve your concerns, we kindly request the reviewer to consider raising the rating. We are happy to address any further questions.

---

> > ### Author Rebuttal · Reviewer_kEMy · 2026-04-02
> >
> > The author's response has addressed most of my concerns, so I will increase score accordingly.

---

> > > ### Author Response · Authors · 2026-04-02
> > >
> > > Thank you very much for your positive feedback! Your thoughtful comments have helped us improve the quality of our work significantly. We truly appreciate your time and effort.

---

### Decision · Program_Chairs · 2026-04-30

**Decision:**

Accept (regular)

**Comment:**

This paper introduces DPPrefSyn, a framework designed to generate differentially private (DP) synthetic preference data for aligning Large Language Models (LLMs). The goal is to address privacy concerns associated with using real human preference data, which often contains sensitive prompts and judgments.

The method operates in three main stages:
- Preference Modeling: It clusters embedding differences of private preference pairs using DP-PCA and DP-KMeans to capture diverse human preferences.
- Reward Learning: It trains per-cluster linear reward models using DP-SGD.
- Data Synthesis: It uses these private reward models to score responses generated from public prompts, creating a reusable, synthetic DP preference dataset.

The concerns about this paper include its limited novelty, its evaluation breadth, and the reward model being too simplistic. I would suggest a "weak accept" for this paper.